# Enhancing quantum teleportation efficacy with noiseless linear amplification

Jie Zhao[1,2], Hao Jeng[1], Lorcán O. Conlon [1], Spyros Tserkis [1], Biveen Shajilal [1], Kui Liu[3], Timothy C. Ralph[4], Syed M. Assad[1] & Ping Koy Lam [1,5] ✉

Quantum teleportation constitutes a fundamental tool for various applications in quantum communication and computation. However, state-of-the-art continuous-variable quantum teleportation is restricted to moderate fidelities and short-distance configurations. This is due to unavoidable experimental imperfections resulting in thermal decoherence during the teleportation process. Here we present a heralded quantum teleporter able to overcome these limitations through noiseless linear amplification. As a result, we report a high fidelity of 92% for teleporting coherent states using a modest level of quantum entanglement. Our teleporter in principle allows nearly complete removal of loss induced onto the input states being transmitted through imperfect quantum channels. We further demonstrate the purification of a displaced thermal state, impossible via conventional deterministic amplification or teleportation approaches. The combination of high-fidelity coherent state teleportation alongside the purification of thermalized input states permits the transmission of quantum states over significantly long distances. These results are of both practical and fundamental significance; overcoming long-standing hurdles en route to highly-efficient continuous-variable quantum teleportation, while also shining new light on applying teleportation to purify quantum systems from thermal noise.

Quantum teleportation is at the heart of many novel quantum technologies. It showcases how one can exploit entanglement to facilitate information tasks that have no classical analogue. Quantum teleportation serves as the elementary ingredient for constructing quantum networks[1,2], distributed quantum computing[3,4], and one-way quantum computation[5–7]. Advances in quantum teleportation, as with other quantum information processing tasks, have followed two directions: encoding information on variables with either discrete or continuous eigen-spectrum. Discrete-variable quantum teleportation has been developed from polarization qubits to qudits employing multiple degrees of freedom of photons[8], and from photonic to hybrid systems[9–13]. Experimental implementations have been extended from the laboratory to intercontinental configurations, and culminated in ground-to-satellite deployment[14].

On the other hand, continuous-variable (CV) quantum teleportation[15] is renowned for its unconditional operations, high compatibility with classical communication infrastructure, and the ease of implementation of most Gaussian operations. Yet, due to the difficulty in generating highly squeezed states and the inevitable decoherence of squeezing, CV teleportation is currently limited to a fidelity of 83%[16]. In

[1]Centre for Quantum Computation and Communication Technology, Department of Quantum Science and Technology, Research School of Physics, The Australian National University, Canberra ACT 2601, Australia. [2]Joint Quantum Institute, National Institute of Standard and Technology and University of Maryland, College park 20742 MD, USA. [3]State key laboratory of quantum optics and quantum optics devices, Institute of Opto-Electronics, Collaborative Innovation Center of Extreme Optics, Shanxi University, 030006 Taiyuan, China. [4]Centre for Quantum Computation and Communication Technology, School of Mathematics and Physics, University of Queensland, St. Lucia QLD 4072, Australia. [5]Institute of Materials Research and Engineering, Agency for Science Technology and Research (A*STAR), 2 Fusionopolis Way, Innovis 138634, Singapore. ✉e-mail: ping.lam@anu.edu.au

principle, unit fidelity is only possible with infinite squeezing, even in a perfect channel. In realistic channels, the current maximum transmission distance is 12 m[17] for an input Shrödinger cat state and table-top distances for coherent and squeezed input states[18–20].

Various approaches have been proposed to overcome these onerous constraints. A recent proposal for high-fidelity CV teleportation[21] relies upon a large network of quantum scissors, each of which acts as a physical noiseless linear amplifier[22–24]. A higher fidelity is only obtainable with increased scales of the network, rendering the experimental demonstration considerably resource intensive. Following a different route, Knill[25,26] and Luo et al.[27] suggested to exploit teleportation to suppress quantum decoherence. The setups they consider however, invariably require quantum error correction.

In this work, through heralded noiseless amplification implemented via the CV Bell measurement, we report on a quantum teleporter that addresses the problems intrinsic to previous CV teleportation protocols. Noiseless linear amplification is capable of amplifying incoming quantum states without degrading the input signal-to-noise ratios[28–32]. Our teleporter inherits this noise-reducing feature, allowing therefore, the recovery of quantum coherence impaired after transmission over long distances. Significant improvements are attainable in fidelity and transmission distance over which teleportation takes place. Using only moderate levels of entanglement resources, unit fidelity is approachable and the purity of pure input states can be preserved. This is in stark contrast to conventional teleportation where infinite energy would be required to achieve the same results.

## Results

### Heralded quantum teleporter

The experimental schematic of our teleporter is depicted in Fig. 1a. Two squeezed single modes with squeezing parameter $r$ are combined to create a two-mode squeezed state, also referred to as the Einstein–Podolsky–Rosen (EPR) state. One arm of the EPR state is coupled to an input state $\rho_{in}$ on a 50:50 beamsplitter and detected on a dual-homodyne station simultaneously measuring the amplitude and phase quadratures. The measurement outcomes, denoted as $\alpha_m = (x_m + iy_m)/2$, undergo a filtering algorithm embodied by an acceptance function

$$f(\alpha_m) \propto \exp\left[\left(|\alpha_m|^2 - \alpha_c^2\right)\left(1 - \frac{1}{g^2}\right)\right] \tag{1}$$

if the quadrature amplitude $|\alpha_m| < \alpha_c$; otherwise, the measurement ensemble is kept with unity probability. In the measurement-based picture, this post-selection, in conjunction with a local rescaling mapping $\alpha_m \to g\alpha$, could effectively emulate a noiseless linear amplifier (NLA) with an amplification gain of $g$[30,33,34]. Its operational regime is determined by the cut-off parameter $\alpha_c$. The approximation to an ideal NLA can be made arbitrarily close to perfect by increasing $\alpha_c$ while retaining a finite success probability. When successful, the heralded amplitude and phase records are rescaled electronically by $\phi_x$ and $\phi_y$, respectively, and broadcast to Bob via a classical communication channel. Bob performs a displacement operation on his EPR mode accordingly to reconstruct the input state.

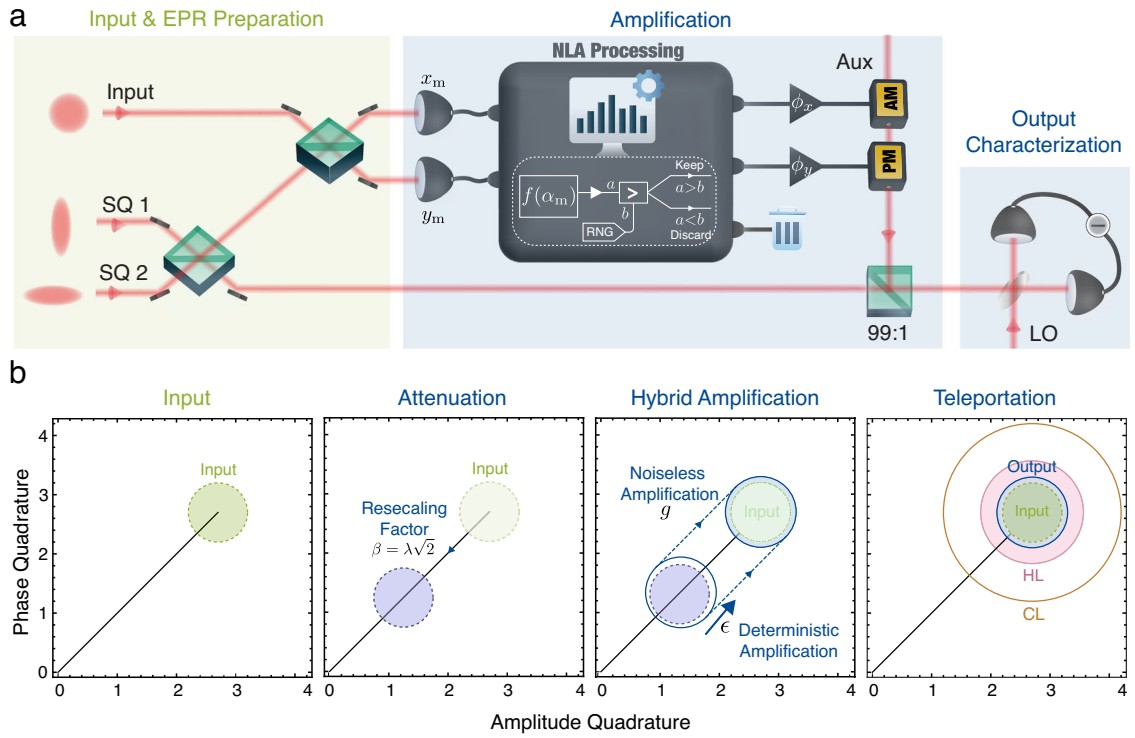

**Fig. 1 | Noise-reduced quantum teleportation. a** Schematic of the optical quantum teleporter. Two squeezed modes are combined to generate the EPR state, which is coupled to the input state. A dual homodyne joint measurement is performed to evaluate both conjugate quadratures. Noiseless amplification, embodied by the NLA Processing panel, is implemented by post-selecting the outcomes of the joint measurement according to an acceptance function $f(\alpha_m)$ (see Methods). The successful events are then amplified by $\phi_{x/y}$ and fedfoward to displace the transmitted EPR mode via a bright auxiliary beam. Lastly, a verification homodyne is employed to characterize the teleported state. **b** Conceptual diagram of the teleporter. The teleportation can be described as a two-step process. First, the electronic rescaling factors $\phi_x$ and $\phi_y$ are set to $\lambda\sqrt{2}$. The transformation is effectively equivalent to a beamsplitter with transmittivity of $\lambda$. Second, deterministic amplification with a gain of $\epsilon$ is applied that inevitably introduces some excess noise. The combined action of $\beta$ and $\epsilon$ is effectively parametrized by $\phi_{x(y)}$, in (**a**) Noiseless amplification with a gain of $g$ is then performed to fulfil the unity-gain condition. The additional noise can be made arbitrarily small by increasing $g$ while reducing $\epsilon$. Note that the teleported state has a significantly reduced noise compared to the optimal results of conventional teleportation schemes using the same entanglement resource, denoted by the Heisenberg limit. RNG random number generator, HL Heisenberg limit, CL classical limit, SQ squeezed beams, AM/PM electro-optic amplitude/phase modulators, LO local oscillator, AUX auxiliary beam.

The working mechanism of the teleporter is conceptually illustrated in Fig. 1b. We first recall the conventional quantum teleportation scheme. In this diagram, the teleportation can be decomposed into a pre-attenuation followed by a post-amplification. The in-loop classical rescaling factor $\phi$ is first tuned to $\beta = \lambda\sqrt{2}$ where $\lambda$ denotes the entanglement parameter $\lambda = \tanh(r)$. Conventionally, this corresponds to the *gain tuning* operational point where the teleporter passively attenuates the input without injecting excess noise. A phase-insensitive linear amplification with gain $\epsilon = 1/\lambda$ is then exploited to satisfy the unity-gain condition, whereby the output has the same quadrature amplitudes as the input. Unity-gain justifies the universality of a quantum teleporter which acts invariantly upon arbitrary input states[18,35]. As elucidated by Haus and Mullen[36] and Caves[37], any phase-insensitive linear amplification is unavoidably subject to a noise penalty equivalent to $|\epsilon^2 - 1|$ units of vacuum noise. The teleported state, therefore, has increased quadrature variances as compared to the input state. It is only possible to avoid this additional noise and hence achieve unity fidelity when $\epsilon \to 1$, that is $r \to \infty$. In contrast, the heralded teleporter incorporates a noiseless linear amplifier to complement the deterministic amplification. The interplay between the two distinctive amplification schemes fulfils the unity-gain condition. The noiseless gain $g$ and the deterministic gain $\epsilon < 1/\lambda$ are tuned with a high premium given to optimizing the output fidelity at a practical success probability. With an increased $g$, thereby decreased $\epsilon$ to conform to unity gain, one can achieve larger noise reduction, and hence higher fidelity enhancement surpassing the quantum-limited performance for a given entanglement resource. This effect is manifested by the quadrature variances of the teleported states:

$$V_{X(Y)}^{\text{out}} = \frac{V_{X(Y)}^{\text{in}} \cosh(2r) + 1}{V_{X(Y)}^{\text{in}} + \cosh(2r)} + \frac{\left(e^{-2r} + V_{X(Y)}^{\text{in}}\right)^2}{g^2\left(V_{X(Y)}^{\text{in}} + \cosh(2r)\right)}. \quad (2)$$

Here, $V_{X(Y)}^{\text{in}}$ denotes the variance of the amplitude (phase) quadrature of the input state. In the limit of $\epsilon$ approaching 1, the post-amplification is fully implemented by the noiseless linear amplification; thus, complete removal of excess noise is approachable, although at the expense of a vanishing success probability. This is achieved with moderate initial squeezings, which would otherwise require an infinite squeezing parameter $r$, so that $\beta = \sqrt{2}$, in conventional deterministic quantum teleportation. Note that the interplay between $g$ and $\epsilon$ promises a continuous operating regime from conventional teleportation to noiseless quantum teleportation with unit fidelity.

Our teleporter exhibits an intriguing feature when operating in the regime of large $g$: given a priori knowledge about the quadrature variances of an input Gaussian state, regardless of the initial squeezing level, $V_X^{\text{out}} V_Y^{\text{out}} = 1$ is approachable for all pure inputs. This preservation of the input purity is only possible with infinite squeezing if conventional quantum teleporter is employed where the output purity parameter is lower bounded by $(2e^{-2r} + 1)^2$. Only if $r \to \infty$, the teleported state remains pure (details provided in Supplementary Note 1). In Supplementary Note 2, we show that our protocol can be applied to purify non-classical input states such as single photon states and Schrödinger cat states.

## Noiseless amplification

We move on to report our experimental results. We first demonstrate noiseless amplification by teleporting a coherent state with real amplitude of $\alpha = 0.11$. The results, as depicted in Fig. 2a, show the output amplitude noise as the mean is amplified. Here we start from the *gain tuning* operational point (the attenuation stage in Fig. 1b) and increase only the noiseless gain $g$ to implement the amplification. Near noiseless amplification is observed compared to conventional quantum teleportation where noise is amplified quadratically.

## Suppression of thermal decoherence

Secondly, we verify the suppression of thermal decoherence where we perform tests with vacuum inputs but maintain the unity-gain condition. As shown in Fig. 2c–e, the variance of the output state can be reduced from 1.73 to 1.16 units of shot noise as the noiseless gain is increased from 1.0 to 1.6, whilst the unity-gain condition is held valid throughout the experiment. This leads to significant enhancement in fidelity that not only surpasses the classical limit, but also the best achievable quantum limit when benchmarked on the same entangled quantum resources (see Fig. 2b and Supplementary Note 1 for detailed derivations). In particular, at $g = 1.6$, we report an output fidelity of $F = 0.92$ with $-6.5$ dB of initial squeezing, exceeding the state-of-the-art teleportation fidelity of 83%[16]. This result would require a squeezed resource of $-13$ dB by means of conventional teleportation subject to the same experimental parameters. All measurements show good agreement with the theoretical curves accounting for experimental imperfections, from which one can extrapolate that unit fidelity is approachable using the present teleportation scheme (details provided in Supplementary Note 3).

Note that the results adhere to the no-go theorem enforced by quantum mechanics on phase-insensitive linear amplifiers[37]. The inevitable noise penalty associated with any linear amplification is embodied here by a decreased success probability accompanying the improvement in fidelity, as illustrated in Fig. 2b. We operate the teleporter at success rates greater than $10^{-5}$. Notably, as addressed in Supplementary Note 1, the success probability tends to decrease more gradually as the noiseless gain continues to increase, indicating that further fidelity growth does not necessarily result in a dramatic decrease in the success probability.

The same suppression of thermal decoherence is observed when teleporting coherent states with displacements in amplitude and phase quadratures: $\alpha = 0.1(i), 0.2(i)$, and $0.4(i)$, as illustrated in Fig. 3. The teleporter demonstrates phase-insensitive and input-invariant behaviour. We observe fidelities, ranging from $F = 0.87 \pm 0.01$ to $F = 0.91 \pm 0.01$, surpassing both the classical and the Heisenberg limits.

## Purification of thermal states

Finally, in Fig. 4, we report a rather intriguing purification effect, which occurs when teleporting displaced thermal states. We consider an input thermal state with coherent amplitude of $|\alpha_{\text{in}}\rangle = |0.23 + 0.01i\rangle$ and quadrature variances of 1.98 in both amplitude and phase. We obtain an output that possesses the same quadrature amplitudes but markedly reduced variances $1.59 \pm 0.1$. The underlying rationale of the thermal suppression is remarkable: the present quantum teleporter takes a mixed quantum state and transforms it into a state of higher purity. This purification would require non-Gaussian operations to achieve conventionally[38,39]. The effect relies on the fact that the NLA amplifies the signal-to-noise ratio of an incident displaced thermal state beyond what it does for a coherent input (see Methods for more details). Since the displacement remains the same, one can ascertain that no coherent signal is contaminated along with the noise removal. This observation showcases a fundamental difference between the present quantum teleporter and conventional methods, which add thermal noise to quantum states but do not remove it.

We note that the purification of quantum states has been discussed in a series of works prior to our observations. Indeed, a recent theoretical proposal by Blandino et al.[34] investigates a teleportation scheme which incorporates post-selection as a critical element. They show that the system is essentially equivalent to a physical noiseless linear amplifier[28], followed by a beamsplitter coupled to a thermal reservoir. Post-selection here effectively transforms the thermal reservoir to a vacuum state; consequently, coherent states can be teleported with less thermal decoherence. Our experimental results with vacuum and coherent states lend direct support to this proposition.

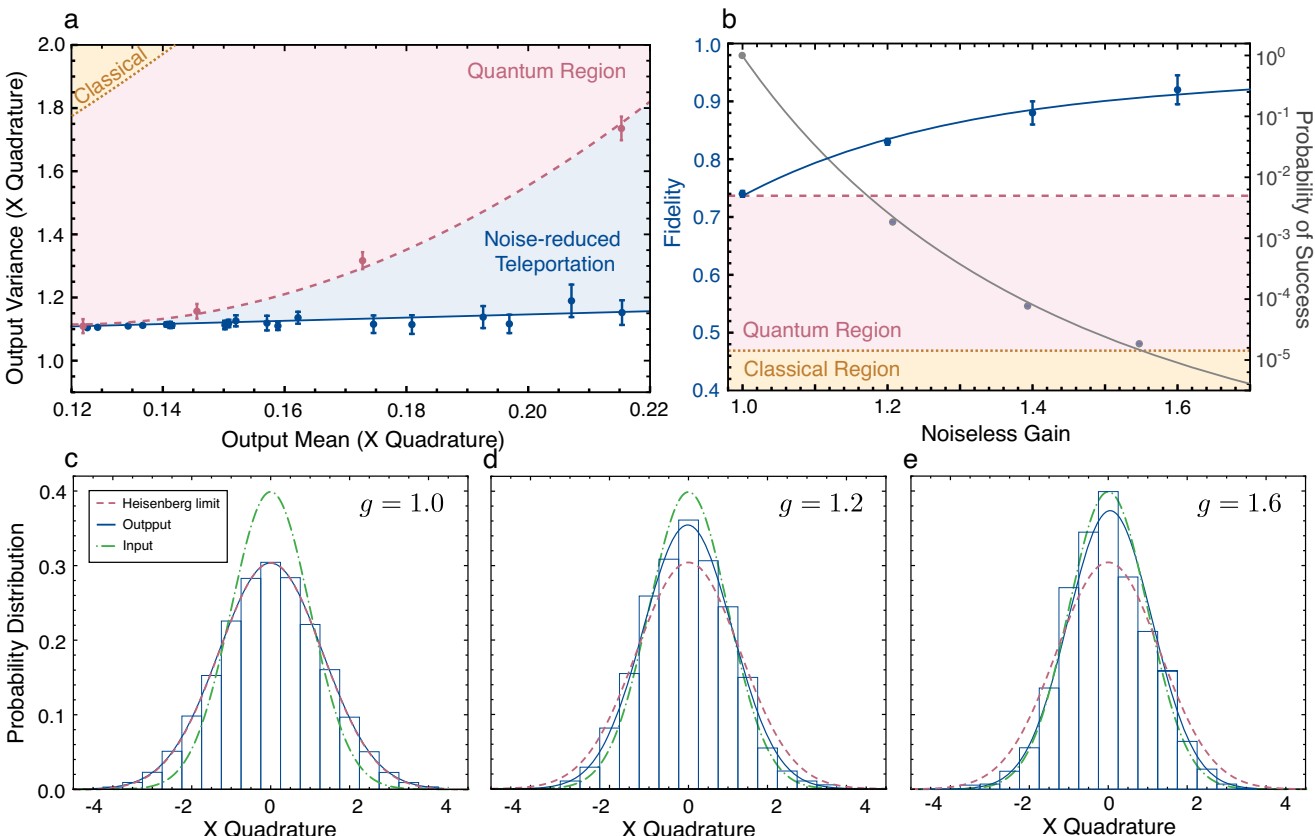

**Fig. 2 | Enhancement in quantum teleportation fidelity. a** Amplitude noise vs mean of the teleported state (blue line), benchmarked against the optimal conventional quantum teleportation (pink dashed curve) and teleportaion without entanglement (brown dotted line). In contrast to the conventional approach, an increase in the output mean is possible with negligible increase in the variance, verifying that our teleporter has a built-in noiseless amplification feature. Error bars represented 1 s.d. of the variance in the amplitude quadrature. **b** Improvement in fidelity over conventional techniques as a function of the noiseless gain (blue). This enhancement comes at the price of finite success probability, as evidenced in the grey curve with labels on the right side. Error bars represented 1 s.d. of the output fidelity. **c**–**e** Histograms of the measured amplitude quadratures of the teleported states for a series of noiseless gains $g = 1.0 - 1.6$. Gaussian fits to the data are given by the blue curves, while the Heisenberg limit (pink dashed curve), and the input (green dash-dotted curve) are superimposed for comparison.

Nevertheless, when this theory is juxtaposed with our observations for displaced thermal inputs, we are led to a quite surprising conclusion that noiseless linear amplification is also capable of suppressing thermal decoherence at its input. This would entail the noiseless amplifier amplifying, and the beamsplitter reducing the displacement and thermal noise in such a way that we win out overall. Even though noiseless linear amplification is known for its ability to noiselessly amplify coherent states and to distill entanglement, the purification effect was not yet discovered. A straightforward calculation shows that the same phenomenon is not attainable using deterministic teleporters or phase-insensitive amplifiers.

The situation described above bears some similarities with the observations of Usuga et al.[40,41], who reported an increase in signal-to-noise ratio of a displaced thermal state following noiseless amplification. However, no increase in purity was demonstrated in their experiments–both the displacement and the noise were increased. Noh et al., on the other hand, have suggested how thermal noise can be removed by a more elaborate quantum error-correction encoding based on the GKP code[42]. However, there the encoding must be implemented before thermalization occurs. Hence, the difficulty lies in the up-front challenge of generating GKP states, which remains a formidable experimental challenge. An averaging technique was proposed by Andersen et al. following the idea of classic entanglement purification[43]. To achieve higher thermal noise suppression, multiple copies of the same input state are required; in contrast, such a stringent requirement is circumvented in our scheme, although we note our teleporter is heralded.

## Discussion
In summary, we propose and experimentally demonstrate a heralded quantum teleporter capable of refining quantum states from thermal decoherence. The enhancement in mitigating decoherence is made possible by trading determinism. We report a significant enhancement in fidelity without demanding more entanglement resources. Near unit fidelity can be achieved with only a modest level of entanglement. In addition, as elucidated in Supplementary Note 4, the teleporter promises near unit fidelity when transmitting quantum states over long-distance channels, however significant the channel loss is. This methodology provides an appealing pathway to obviate the bottleneck of CV quantum teleportation, enabling teleportation between truly remote end users.

Our work opens up a number of interesting future directions. High-fidelity quantum teleportation, in its own right, has distinct merits in a broad class of quantum information and computation applications[44,45]. Notably, in the Supplementary Note 5, we show that our heralded teleporter when applied to EPR entanglement results in improved quantum correlations successfully shared between the end users. In this regard, the teleporter works as an entanglement swapping protocol that outperforms conventional deterministic swapping schemes[46,47]. This provides a feasible pathway to reducing the resource overhead in constructing regenerative relays and hence quantum

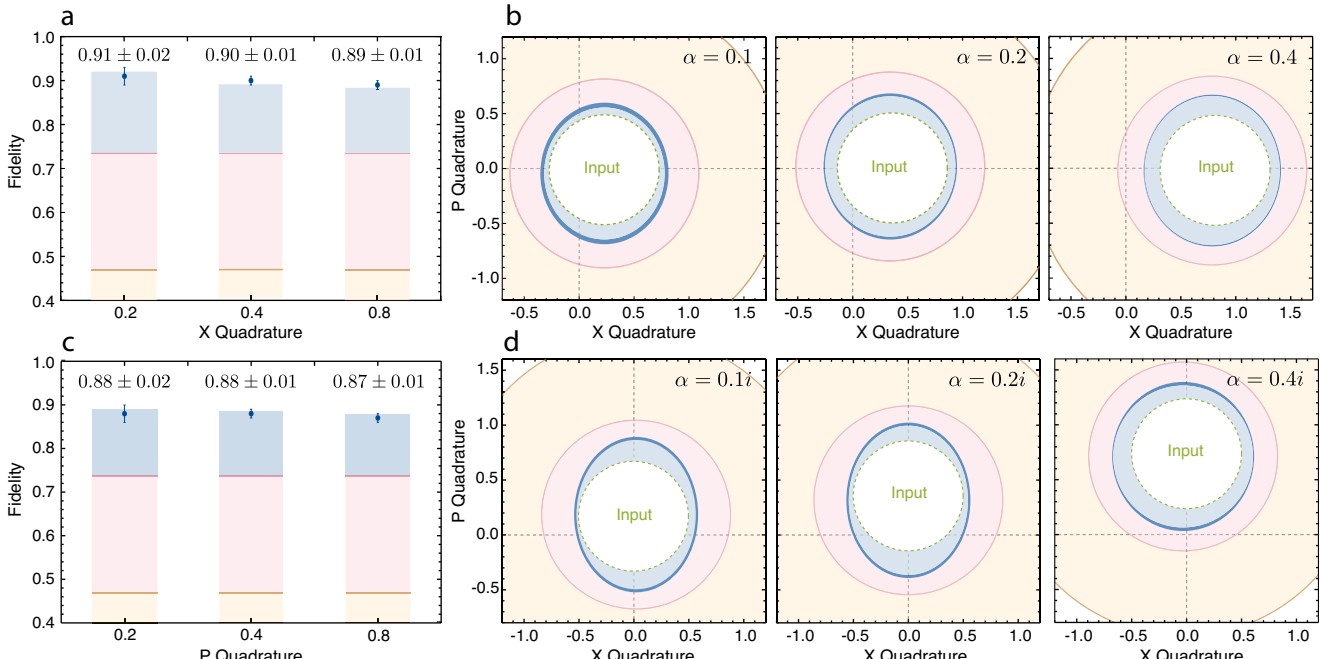

**Fig. 3 | Quantum teleportation of coherent states. a, c** Recorded fidelities for various coherent states that are displaced in amplitude and phase quadratures, respectively. Error bars represent 1 s.d. of the output fidelity. Bars refer to the theoretical expectations of the output fidelities (blue), together with the corresponding Heisenberg limit (pink) and classical limit (brown). **b, d** Teleported states depicted in phase space. Inner circle (green dashed) represents the input state, blue shaded region denotes the operational regime of our noise-reduced teleporter, whilst the pink and brown shaded regions show the accessible working spaces of the conventional and classical teleporters, respectively. Darker blue bands represent the uncertainties associated with the estimation of variances in output quadrature amplitudes. Contours: 1 s.d. width of the corresponding Wigner functions.

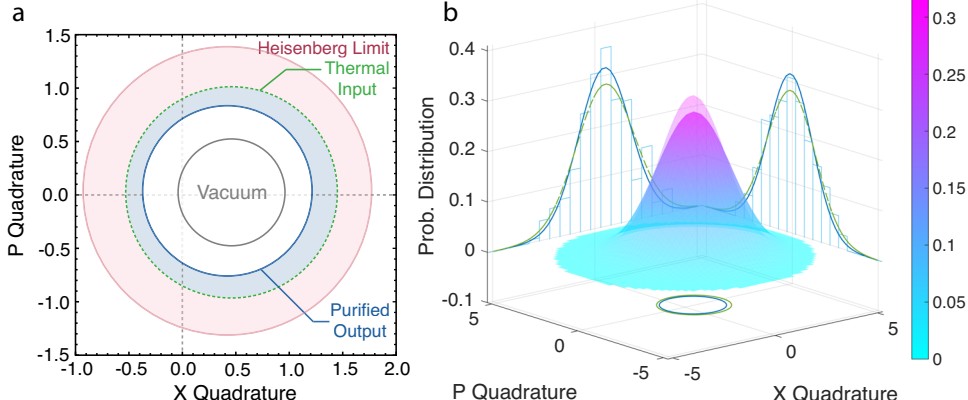

**Fig. 4 | Purification effect. a** Purification of a displaced thermal state that reduces the input noise while preserving the input displacement. In contrast, a conventional teleporter would introduce additional noise. **b** Wigner functions of the input and output (semi-transparent) together with the corresponding quadrature probability distributions (green for the input and blue for the output). Quadrature measurement histograms of the output state that are constructed from $5 \times 10^8$ measurements are superimposed.

repeaters when highly squeezed quantum resources are not available[48–51]. In distributed quantum computing[4,52,53] and sensing[54,55], and quantum key distribution[56,57] where resources are prepared offline, trading determinism for higher fidelity and enhanced entanglement distribution can be beneficial. In this work, we have restricted ourselves to operate under the unity-gain condition, so that our teleporter acts invariantly on all input states. In future, by relaxing this condition, our teleporter can be adopted to simulate a variety of quantum channels[34,58–61] otherwise implausible using deterministic setups. The protocol opens up avenues such as quantum tele-amplification and tele-cloning with fidelity surpassing the no-cloning limit[62–65], that have been proven useful in coherent-state quantum computing[66,67], but are

inconceivable deterministically. Furthermore, our approach demonstrates an intriguing alternative for mitigating channel loss without physically introducing non-Gaussian operations[34,68–70]. As an application, the protocol enables one to distill a Bell state sufficiently to violate the Clauser, Horn, Shimony, Holt (CHSH) inequality. Each of the above possibilities merit investigations and could be of interest from either practical or fundamental perspectives.

## Methods
### Experimental arrangement
The primary light source is a 1064 nm infrared laser (Innolight Dioblo). The laser is frequency-doubled via internal parametric up-conversion

to provide both the fundamental seed beams and second-harmonic pump beams used to drive a pair of identical optical parametric amplifiers (OPA). The OPAs are configured to be bowtie-shaped ring cavities approximately 30 cm long in round-trip length, each hosting a single PPKTP crystal. Intra-cavity optics are HR coated on the front surfaces to enable double resonance at both 1064 nm and 532 nm, with the crystal temperature adjusted to maximize the parametric gain. Active servo-lockings are performed in real time to control (1) the cavity length to ensure double-resonance is always satisfied, and (2) the relative phase between the pump and the seed beams to ensure that the OPAs operate at de-amplification, giving rise to amplitude squeezing. Error signals are generated using a Pound-Drever-Hall-like method with amplitude and phase modulations. The incident pump powers and parametric gains of the two OPAs are carefully tuned such that the squeezed vacuums are of identical squeezing levels before being combined to yield an EPR state. The OPAs constitute the main workhorses of the experiment that are capable of generating up to 11.5 dB of vacuum squeezing. Squeezing of 7.3 dB with purity of 1.17 was observed. The round-trip loss is estimated to be between 0.5% and 1%.

Quantum states are defined to be photons residing at 3.12 MHz frequency sidebands of the electromagnetic carrier field, created through amplitude and phase electro-optical modulations. The output photons of interest are characterized via demodulation at 3.12 MHz with a bandwidth of 100 KHz. Electro-optic feed-forward is implemented with analogue amplifiers, attenuators, bandpass filters, and coaxial cables. The group delay is 160 ns at 3.12 MHz. The electronic gain is tailored by means of step-attenuators with 0.1 dB step resolution. The in-loop dual homodyne measurement employs a pair of current-subtraction detectors with optimized dark noise clearance, i.e. 20 dB, near 3 MHz in order to minimize excess noise coupled into the output via feed-forward. Verification measurements on the teleported states are made with a voltage-subtraction detector that can be locked to either the amplitude or the phase quadratures, with about 15 dB dark noise clearance.

The total transmission coefficient throughout the setup is estimated to be ~90%, accounting for impure squeezed resources, optical propagation loss, homodyne detection efficiency, and imperfect displacement operations. The theoretical model taking into account of experimental imperfections is constructed based on these loss parameters, as detailed in Supplementary Note 3.

## Post-selection

A pair of measurement outcomes $(x, p)$ from the in-loop joint measurement is post-selected with acceptance probability proportional to $P(x,p) \propto \exp(A(x^2 + p^2))$. For practical purposes, the coefficient $A$ is parametrized by the "noiseless gain" $g$ as

$$A = \frac{1}{2}\left(1 - \frac{1}{g^2}\right).$$

With $g > 1$, $A$ is positive so that $P$ is divergent. We thus have to "cut it off" at large values, and normalize the maximum value to 1 to allow an interpretation in terms of probabilities. The function $P$ therefore takes the form

$$P(x,p) = \begin{cases} P_0 \exp(A(x^2 + p^2)), & x^2 + p^2 \leq B, \\ 1, & x^2 + p^2 > B. \end{cases} \quad (3)$$

where $P_0 = \exp(-AB)$. The variable $B$ is the cut-off, chosen to be large enough to ensure that the output state remains Gaussian, or, equivalently, that the noiseless amplification is linear[33]. In our experiments, $B$ has a fixed value of 4.7.

In practice, for each teleportation run, $5 \times 10^8$ measurements are recorded simultaneously at the in-loop dual homodyne and the verification homodyne stations, where the verification homodyne is locked to amplitude and phase quadratures consecutively. Output statistics is post-selected by exploiting covariances of the joint distribution, enacting effectively the noiseless amplification.

## Displaced thermal input states

We discuss in more details about the purification effect and provide a way to visualize the basic physics underlying the action of the teleporter on a displaced thermal input state. Recall the scheme shown in Fig. 1, we consider the situation where the amplification is fully implemented by the NLA, which means the deterministic gain $\epsilon \to 1$. The scheme can be visualized as a concatenation of a beamsplitter with transmittivity $\tanh(r)$ and an NLA with noiseless gain $g$. Here we have a displaced thermal state of mean amplitude $\alpha_0$ and parameter of $\lambda_0$ where $\lambda_0^2 = (V_0 - 1)/(V_0 + 1)$ with $V_0$ being the variance. The input state can be expressed as

$$\rho_{in} = D(\alpha_0)\rho_{in}(\lambda_0)D^\dagger(\alpha_0), \quad (4)$$

where $D(\alpha_0)$ is the displacement operator. P function of the input thermal state is given by

$$\int d^2\alpha \frac{1}{\sqrt{\pi}} \sqrt{\frac{1 - \lambda_0^2}{\lambda_0^2}} e^{-\frac{(\alpha - \alpha_0)^2}{\lambda_0^2}(1 - \lambda_0)^2} |\alpha\rangle\langle\alpha|. \quad (5)$$

The beamsplitter simply transforms the input state to another displaced thermal state

$$\rho_{BS} = D(\alpha_0')\rho_{in}(\lambda_0')D^\dagger(\alpha_0'), \quad (6)$$

where

$$\begin{aligned} \alpha_0' &= \tanh(r)\alpha_0, \\ \lambda_0' &= (V_{BS} - 1)/(V_{BS} + 1), \quad \text{and}, \\ V_{BS} &= \tanh^2(r)V_0 + 1 - \tanh^2(r). \end{aligned} \quad (7)$$

The action of the noiseless linear amplifier on $\rho_{BS}$ produces an output state with P function

$$\begin{aligned} \int dx dy \frac{1}{\pi} \frac{1 - \lambda_0'^2}{\lambda_0'^2} e^{-\frac{1 - \lambda_0'^2}{\lambda_0'^2}[(x - x_0')^2 + (y - y_0')^2]}, \\ e^{|\alpha_0'|^2(g^2 - 1)} |g\alpha_0'\rangle\langle g\alpha_0'|. \end{aligned} \quad (8)$$

Arranging Eq. (8) into a more succinct form reveals that the output state is in fact a displaced thermal state with mean amplitude and variance of

$$\begin{aligned} \alpha_{out} &= g \frac{1 - \lambda_0'^2}{1 - g^2\lambda_0'^2}\alpha_0' = \frac{2g\alpha_0'}{1 - g^2(V_{BS} - 1 + V_{BS})}, \\ \lambda_{out} &= g\lambda_0' = (V_{out} - 1)/(V_{out} + 1), \quad \text{and}, \\ V_{out} &= \frac{1 + g^2(V_{BS} - 1) + V_{BS}}{1 - g^2(V_{BS} - 1) + V_{BS}}. \end{aligned} \quad (9)$$

This means for an incident displaced thermal state ($\rho_{BS}$ here), the NLA amplifies its mean amplitude by an effective gain

$$g_{eff} = g \frac{1 - \lambda_0'^2}{1 - g^2\lambda_0'^2} = \frac{2g}{1 - g^2(V_{BS} - 1 + V_{BS})} > g, \quad (10)$$

since $g\lambda_0'$ must remain smaller than 1 for the amplified state to be physical. Consequently, an NLA amplifies a thermal state slightly more than it does to a coherent input state. This fact is essential for realizing

the purification effect. For all $1 < g < g_{max} = 1/\lambda_0'$, the signal-to-noise ratio (SNR) of the output is always greater than that of the input state.

To satisfy the unity-gain condition, the input and output mean amplitude must be equal, i.e. $\alpha_{out} = \alpha_0$. As a result of this condition, one obtains the relation between $g$ and $r$ given by

$$g = \frac{\sqrt{V_0^2 - 1 + \tanh^2(r)} - \tanh(r)}{V_0 - 1}. \tag{11}$$

Substituting Eq. (11) into Eq. (9), one obtains the overall output variance that has $V_{out} \leq V_0$. The rationale behind is that by concatenating an NLA to a beamsplitter (with no excess noise), we can retain the input mean but obtain an output with reduced thermal noise. If we relax the unity-gain condition, by setting $g = g_{max} = 1/\lambda_0'$, the output becomes a pure coherent state.

The results relate to the proposal by Blandino et al. in refs. 34,69, where they characterize the effective quantum channel established by a teleporter embedded with a measurement-based NLA. The transformation enacted by the protocol is equivalent to an effective system comprised of a noiseless amplifier (or attenuator), followed by a quantum channel which can, in principle, have arbitrarily small loss and coupled thermal noise. The results coincide with this picture. Owing to the channel purification effect, the protocol can be employed to probabilistically purify quantum non-Gaussian states using only Gaussian operations.

## Data availability

The data generated in this study have been deposited in the figshare database (hyperlinks: https://doi.org/10.6084/m9.figshare.23702427.v1, https://doi.org/10.6084/m9.figshare.23702262.v1, https://doi.org/10.6084/m9.figshare.23702145.v1).

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

## Acknowledgements

The authors acknowledge support from the Australian Research Council (ARC) via the Centre of Excellence for Quantum Computation and Communication Technology (CE110001027). P.K.L is an ARC Laureate Fellow. K.L. would like to acknowledge the National Natural Science Foundation of China (Grant No. 12074233).

## Author contributions

J.Z., J.H., L.C., K.L., S.A. and P.K. designed and built the experiment. J.Z., J.H., S.A., S.T. and T.C.R. conceived and developed the theory. J.H., J.Z., B.S. and L.C. collected and analysed the experimental data. All authors discussed the results, wrote and commented on the manuscript.

## Competing interests

The authors declare no competing interests.
