## [Peer Review File · Nature Communications]

Enhancing quantum teleportation efficacy with noiseless linear amplificationREVIEWER COMMENTS

Reviewer #1 (Remarks to the Author):

The authors investigate and experimentally demonstrate a continuous-variable quantum teleportation protocol whose performance is enhanced by conditional noiseless amplification. Intriguingly, the noiseless amplification is accomplished by rejecting or accepting the teleportation attempt depending on the outcomes of homodyne detections on Alice's side of the teleportation protocol. This filtering is based on an inverse Gaussian acceptance function (up to certain cut-off threshold), which can emulate the noiseless amplification in eight-port homodyne detection of the amplified state. The reported improvement of CV quantum teleportation can potentially have large impact on quantum technologies, but there are several important issues and questions that should be addressed by the authors:

Let us first consider teleportation of coherent states. When Alice and Bob share two-mode squeezed vacuum state with finite squeezing, Alice's measurement reveals some information about the teleported state that is supplied by Victor. From local Alice's perspective, she attempts to estimate the complex amplitude of Victor's coherent state but the measurement is blurred by thermal state injected into one port of the balanced beam splitter. If Alice applies the inverse Gaussian filter to her data, then it seems that she can estimate the amplitude of the coherent states with higher precision, because the signal to noise ratio of her measurement conditionally improves, at the expense of rejecting most of the data. This can be seen as an instance of probabilistic state discrimination. Recall that two coherent states with opposite amplitudes can be probabilistically discriminated by homodyne detection with arbitrary precision if one accepts only the measurement results where the absolute value of the quadrature exceeds some (large enough) threshold. The present protocol seems to make use of a more elaborate version of this basic scheme. Note that Eq. (24) predicts that in the limit of large gain, teleportation with fidelity arbitrarily close to unity is achievable even for $r=0$, i.e. without any shared entanglement. In that case one gets a measure-and-prepare protocol, where Alice's filtering helps to estimate the amplitude of coherent state with high precision, before it is sent to Bob. Of course, in practice this could only work for some limited range of amplitudes, since the cut-off has to

be finite.

If Alice learns some classical information about the teleported state, then the state transfer cannot be perfect and universal and the teleportation channel should introduce some decoherence. The authors should theoretically analyze how their scheme would work for teleportation of coherent superpositions of coherent states (Schrodinger cat-like states) or whether it would be useful for entanglement swapping, i.e. teleportation of one part of two-mode squeezed state. More generally, it would be very useful to specify the effective quantum channel established by the considered probabilistic teleportation protocol.

The authors also report a purification effect that their protocol has on input thermal states. My suspicion is that this purification effect is connected with the cut-off in the filtering and with the fact that coherent states with large amplitudes are transferred with less-than-unity gain. A thermal state can be seen as a Gaussian mixture of coherent states. If every coherent state would be teleported perfectly and with unity gain, then also the thermal state would be teleported perfectly. In the considered probabilistic teleportation protocol, the success probability of coherent state teleportation increases with the absolute value of the complex amplitude (up to some cut off). This should in fact lead to increase of thermal noise, since the Gaussian P-function of the thermal state would be multiplied by an inverse Gaussian. However, if the noise is large enough, then the coherent states with large enough amplitude are transferred with less-than-unity gain due to cut-off in filtering, and the thermal noise is effectively reduced. It is then questionable what would be the applicability of such purification effect and the authors should discuss in more detail in the manuscript its physical origin to clarify this point.

Finally, the authors find that in the large gain limit pure input squeezed states can be teleported onto pure states. The authors should clarify whether this requires the knowledge of the quadrature variances of the teleported state and whether the classical gains in the teleportation protocol need to be adjusted according to these quadrature variances, as implied by Eq. (19). Dependence on the input quadrature variances makes the protocol less general. Note also that the pure-state teleportation can be approached if one conditions on measurement outcomes close to zero, i.e. postselects $x_m=y_m=0$. This has been

demonstrated by Akira Furusawa group. It would be worth to compare the present approach with the postselection on x_m and y_m being close to 0, to see what are the trade-offs between the purity and the success probability of the protocol.

Reviewer #2 (Remarks to the Author):

The submitted manuscript presents an experimental realization of a teleportation protocol that utilizes noiseless amplification to improve the fidelity of the transmitted coherent states at the cost of success rate. The amplification is based on probabilistic post-selection of the results of the bell-type measurement. The obtained experimental results demonstrate that coherent states can be indeed teleported with a reduced amount of added thermal noise and improved fidelity.

The article then claims to demonstrate a 'near unit fidelity when transmitting quantum states over long distance channels, however significant the channel loss is.' However, this is demonstrated only by using coherent states, which are both Gaussian and classical, and solely described by their first moments of quadrature operators. As a consequence, there are protocols that work on coherent states, such as those in references [41, 42], that are not suitable for general quantum states required for quantum computing tasks. The theoretical discussion of the protocol is also at the level of moments of quadrature operators and therefore also valid for coherent (or Gaussian states). It is not clear whether the unity gain regime works for different quantum states. It is also not clear, whether the purification effect is obtained for other than coherent states, or whether it works for all coherent states or just a subset. This also means that the claim in the conclusion about using the proposed amplifier as a quantum repeater is not supported by the rest of the manuscript.

In summary, my main issue is that the manuscript does not discuss the limitations of the setup. The performance of the setup is demonstrated on several coherent states but this fact and its consequences are glossed over in the discussion. If the protocol can be used also for non-classical and non-Gaussian quantum states, discussing and demonstrating this would make the article suitable for publication in Nature communication. On the other hand, if the protocol only works for coherent states, I would recommend a different and more specialized journal.

Reviewer #3 (Remarks to the Author):

Jie Zhao and colleagues present an interesting spin on the traditional continuous-variable teleportation protocol that can attain higher than usual fidelities at the expense of probabilistic operation. In fact, it can approach unit fidelity, even for realistic squeezing levels. The scheme is demonstrated experimentally for teleportation of coherent states and thermal states, and also includes teleamplification where the output coherent state amplitudes are boosted.

The protocol makes use of a variant of the virtual noiseless linear amplifier (NLA) that was developed by some of the authors (and others) and implemented by them for several different applications. The virtual NLA works by filtering or post-selecting measurement outcomes and is much more experimentally feasible than a physical NLA. The experimental scheme employed here is essentially the good old CV teleporter constructed from two-mode squeezed light, a Bell (dual homodyne) measurement, and active feed-forward. However, now the teleported states are only kept when the intermediate Bell measurement outcomes pass a certain filtering test which accepts high-amplitude outcomes with higher probability than outcomes close to the origin of phase space. This filtering effectively biases the statistics of the post-selected output states towards larger amplitudes, hence acting as an amplifier. Since this amplification is noiseless (made possible by the probabilistic/heralded operation) and takes over from part of the feed-forward gain applied in the standard teleportation protocol, the teleported states suffer less from the noise added due to finite squeezing of the entangled resource state. Interestingly, the protocol can also act as a purifier of input thermal states, basically through the attenuation + noiseless amplification process.

Since the protocol is heralded - in fact, success probabilities are rather low - it will have reduced applicability compared to standard CV teleportation which is famously deterministic. Still, it will surely be of interest to many in the community, especially with the record-high teleportation fidelities and the flexibility in its operation. The experimental results are convincing, and the paper contains a good discussion of the protocol's relation to similar purification schemes previously proposed or demonstrated.

However, the presentation is let down by somewhat sloppy writing. Most glaringly, the working principles of the protocol and the actual experimental implementation are very poorly explained. It was essentially impossible for me to understand this without studying the supplement and some of the group's earlier papers - even though I was roughly familiar with those prior works. While other readers will certainly be brighter than me, I am sure I will not be the only one left baffled by the central section explaining the protocol - essentially page 2, left column. Here are some of the problems with this passage:

- * The authors seem to assume that the reader is intricately familiar with previous works on virtual noiseless amplification (like they themselves of course are).
- * There is essentially zero explanation of the experimental scheme (Fig. 1a). All discussion is concerned with the dynamics illustrated in Fig. 1b. The description of the experiment is left for the figure caption and the methods section.
- * Because of this, the text "where λ denotes the entanglement parameter [...] and r being the squeezing parameter" is hanging in the air, referring to nothing previously mentioned.
- * Similarly, there is no mention of the central element of the scheme, namely the post-processing filter. Due to this (and probably other missing explanations), it does not become clear how the noiseless amplification actually is performed. I also found that it would be useful to have a bit more detail in the NLA processing panel in Fig. 1 - perhaps including details from Fig. 5 in Methods and Fig. 1 in the appendix.
- * It is unclear what is meant by "an arbitrary unknown input state is attenuated with an electronic rescaling factor". This refers to a scaling of the measurement outcomes, but sounds like actual attenuation of the input state, especially since nothing has yet been said about the data processing.
- * The sentence beginning "By virtue of ϵ being arbitrarily close to unity" does not make sense to me, even after re-reading the paper and the sentence several times.
- * It is unclear whether the scheme works for truly arbitrary input states, only for Gaussian states, or only for coherent states. The paragraph starts with "arbitrary unknown input state" but ends with an expression for the output variances, which I assume is only valid for the Gaussian states that are treated in the supplement.
- * The relation between the many different parameters (ϵ , g , β , $\phi_{x,y}$) and their meaning is

somewhat opaque, especially since there are basically no formulas in the main text.

* "...complete removal of excess noise is obtained at the expense of a vanishing success probability" - this is the first time that it is explicitly mentioned that the protocol is probabilistic (apart from the word "heralded"). It is not explained why it is so.

I encourage the authors to rewrite this section, putting on the glasses of someone who understands standard CV teleportation but is perhaps unfamiliar with NLAs and in particular the virtual kind. If length becomes an issue, some of the introduction or discussion could perhaps be sacrificed, or the figure size or caption text be reduced. I find it essential to the appreciation of this paper that the protocol is properly explained. It might also be useful to refer to their earlier work (or other references) at appropriate points in this section.

A few other points for consideration:

- * A brief discussion of possible scenarios where the scheme would be beneficial in spite of its probabilistic nature would be instructive.
- * The short paragraph describing the teleamplification result appears abruptly without much introduction or connection to the previous text.
- * Section 1 of the appendix could benefit from some polishing:
- * Some steps are not explained well, e.g. 2nd equality of eq. (6), eq. (10), eq. (16).
- * In relation to eq. (2), a description of the initial squeezed input states would be in place.
- * The notation for correlations is all over the place, using Δ , δ , implied Δ , and V in one big mixture.
- * Is it possible to write an explicit expression for $\phi_x(y)$?
- * $\langle \rangle$ missing around $Y_1 Y_1$ in (7).
- * The tilde-correlation appears to be on the wrong side of the fraction in the Y -expression, first line of eq. (14).
- * Prime that shouldn't be there in (13).
- * The mode order used (stated in (5)) is not so helpful.

// Jonas Neergaard-Nielsen

Reviewer #1

We thank the reviewer for his/her time reviewing our manuscript. The reviewer suggests many interesting generalizations of our work and we performed new investigations accordingly to address the comments. Two new sections were added as a result in the Supplementary materials and one section was added in Methods of the main text. We believe these major changes prompted by the reviewer has improved the clarity and quality of our work, and hope the reviewer will allow a recommendation for publication.

The authors investigate and experimentally demonstrate a continuous-variable quantum teleportation protocol whose performance is enhanced by conditional noiseless amplification. Intriguingly, the noiseless amplification is accomplished by rejecting or accepting the teleportation attempt depending on the outcomes of homodyne detections on Alice's side of the teleportation protocol. This filtering is based on an inverse Gaussian acceptance function (up to certain cut-off threshold), which can emulate the noiseless amplification in eight-port homodyne detection of the amplified state. The reported improvement of CV quantum teleportation can potentially have large impact on quantum technologies.

We are happy that the reviewer found the noiseless amplification intriguing and that our work would potentially have large impact on quantum technologies.

But there are several important issues and questions that should be addressed by the authors:

1) Let us first consider teleportation of coherent states. When Alice and Bob share two-mode squeezed vacuum state with finite squeezing, Alice's measurement reveals some information about the teleported state that is supplied by Victor. From local Alice's perspective, she attempts to estimate the complex amplitude of Victor's coherent state but the measurement is blurred by thermal state injected into one port of the balanced beam splitter. If Alice applies the inverse Gaussian filter to her data, then it seems that she can estimate the amplitude of the coherent states with higher precision, because the signal to noise ratio of her measurement conditionally improves, at the expense of rejecting most of the data. This can be seen as an instance of probabilistic state discrimination. Recall that two coherent states with opposite amplitudes can be probabilistically discriminated by homodyne detection with arbitrary precision if one accepts only the measurement results where the absolute value of the quadrature exceeds some (large enough) threshold. The present protocol seems to make use of a more elaborate version of this basic scheme. Note that Eq. (24) predicts that in the limit of large gain, teleportation with fidelity arbitrarily close to unity is achievable even for $r=0$, i.e. without any shared entanglement. In that case one gets a measure-and-prepare protocol, where Alice's filtering helps to estimate the amplitude of coherent state with high precision, before it is sent to Bob. Of course, in practice this could only work for some limited range of amplitudes, since the cut-off has to be finite. If Alice learns some classical information about the teleported state, then the state transfer cannot be perfect and universal and the teleportation channel should introduce some decoherence.

The comments raise an interesting perspective. The reviewer is correct that the noiseless amplification can be employed to improve the probability of successful state discriminations. We studied this topic and reported the results in [PRA 96, 012319 and Int. J. Quantum Inform. 15 1750009]. Consider that Alice measures an unknown input state, by adopting a measure-and-prepare approach, she can evaluate the state with arbitrarily high precision, at the expense of a finite success probability. Conditioning upon a successful event, Alice broadcasts the measurement outcomes via classical channel, and Bob prepares the state accordingly. In principle, the input state can be reconstructed nearly perfectly. This is the regime pointed out by the reviewer whereby unity fidelity is achievable even for $r = 0$, i.e. without any shared entanglement. However, we would like to emphasize that there are significant distinctions between our

heralded quantum teleportation and the classical approach described above. We address the distinctive differences from two perspectives:

- 1) The measure-and-prepare scheme cannot effectively transfer non-classical states such as the EPR states. In contrast, as elaborated in the new Supplementary section V, applying our protocol to teleport EPR entanglement could result in improved quantum correlations between the two sub-modes owned by Alice and Bob. The results constitute an entanglement swapping protocol having improved performance than the conventional schemes [PRA 61, 010302, PRA 60, 2572, PRA 68,012319], when benchmarked on the same entangled resources.
- 2) The significance of using quantum teleportation is to securely share quantum information between remotely-located end users. In the measure-and-prepare picture, Alice broadcasts all the information about the input states via classical channels, implying that Bob and Eavesdropper have the same amount of information about the input states. In this regard, the EPR channel for teleportation is necessary.

The authors should theoretically analyze how their scheme would work for teleportation of coherent superpositions of coherent states (Schrodinger cat-like states) or whether it would be useful for entanglement swapping, i.e. teleportation of one part of two-mode squeezed state.

We thank the reviewer for the suggestions. Each of these possibilities merit thorough investigations. We performed some preliminary investigations and added two new Supplementary sections to report the results. We outline the results in the following.

- 1) In Sec. II, we generalize our previous analysis that is based on the covariance matrix method. Instead, we provide a general formalism to analyze our protocol using Wigner functions. This way, we can study the input and output states' other quantum properties such as negativity rather than just their 1st and 2nd-order moments. We investigate a Schrödinger's cat state as input, according to the reviewer's comment, as well as a single photon input state. In both circumstances, the inverse Gaussian filter can be optimized to obtain a higher fidelity with success probability above 10^{-3} .

More specifically, Fig. S5 plots the output Wigner functions for a single photon input (left) and a cat input (b). We acknowledge the importance of preserving the negativity of these non-classical states [Nat 566 509, Nat Nanotech 12 1026, Science 370, 1460, Nat Photon 5 52]. As evidenced in Fig. S6, our teleporter shows improved performance in retaining the negativity of these states. In particular, the negativity of the single photon state is almost intact which would require infinite squeezing conventionally.

In light of the work reported in [PRL 113, 223602; PRL 109, 180503; Nat Photonics 9, 764], we notice that our scheme with $g < 1$ in the filter function $f(\alpha_m) \propto \exp[(|\alpha_m|^2 - \alpha_c^2)(1 - 1/g^2)]$ effectively approximates a noiseless linear attenuator which was proven to be useful in preserving the purity of these non-Gaussian and non-classical states. We thus operated our teleporter with $g < 1$ and obtained fidelities and success probabilities of $F = 0.9976, P_s = 0.0085$ for a cat input and $F = 0.8732, P_s = 0.0034$ for an input single photon state, respectively. These results are similar to the ones obtained using the inverse Gaussian filter, as shown in Fig. S5. However, the significance of these investigations lies in illuminating the more general applicability of our scheme – its operational regime should not be restricted to $g > 1$, but instead to be appropriately tailored relative to the input alphabet.

Fig. S5. F and P_s are the output fidelity and success probability of the heralded quantum teleporter, respectively. F_{DET} is the obtainable fidelity using deterministic teleporters, when benchmarked on the same entangled resources.

Fig. S6. Wigner functions $W(0,y)$ of the input and output states from our heralded quantum teleporter and the deterministic teleporter.

- 2) In response to the reviewer's comment, we added a new section (Sec. V) in the Supplementary Material to address our investigations on entanglement swapping, *i.e.* teleportation of one part of a two-mode squeezed state (EPR state). We evaluated the performance of the swapping protocol using two figures of merits that haven't been used previously in deterministic swapping protocols: one being an inequality adopted by Tan *et al.* [PRA 60, 2752] $\langle \Delta(X_A - X_B)^2 \rangle + \langle \Delta(Y_A + Y_B)^2 \rangle < 4$ which is a sufficient condition for entanglement and another being *entanglement inseparability* proposed by Duan *et al.* [PRL 60, 2752] which is a necessary and sufficient condition for entanglement.

Regarding the first entanglement witness, thanks to the noiseless linear amplifier, the inequality can be saturated with less initial EPR squeezing, as compared to a deterministic scheme [PRA 61, 010302, PRA 60, 2752, PRA 68, 012319]. Interestingly, we notice that there exists an operational

regime where the conditional variance equals $4 e^{-2r_1}$ where r_1 is the squeezing parameter of the EPR state to be teleported. This is achievable with moderate entangled resources, which would otherwise require infinite squeezing using the deterministic swapping schemes.

In addition, the heralded teleporter results in improved inseparability as compared to the conventional deterministic schemes, when operating under both unity-gain and non-unity-gain conditions.

Fig. S10. Entanglement inseparability of the EPR sub-modes successfully shared between Alice and Bob. (a) operating the heralded teleporter in the non-unity gain regime, and (b) under the unity-gain condition. r_2 is the squeezing parameter of the EPR channel in teleportation.

These results open up new avenues for constructing quantum repeaters with reduced resource overhead. Entanglement swapping can be implemented efficiently when highly squeezed quantum resources are not available. In the circumstances discussed above, the noiseless gains are not optimized. It would be interesting to derive explicit expressions of the optimal noiseless gain and the limitation on the gain values. It is also worthwhile adopting other entanglement measures to assess the protocol. We leave these subjects to future investigations.

More generally, it would be very useful to specify the effective quantum channel established by the considered probabilistic teleportation protocol.

This is an insightful suggestion. Fidelity certainly provides a useful measure for the efficiency of teleportation with respect to the resemblance between the input and output and therefore is the most widely used criteria for CV quantum teleportation [J. Mod. Opt. 47, 267]. But it does not reveal the exact information transfer coefficient during the teleportation process. For instances where the input states are non-classical, as studied above, the ability to preserve the quantum feature of the states would constitute a more effective measure. Besides, there might be situations where the input and output can be interconverted by reversible transformations independent of the input. To this end, other criteria, such as equivalent input noise [PRA 64, 010301], the T-V diagram [PRL 81, 5668] in analogy to quantum non-demolition measurement [Nat 396, 537], and the ν - τ diagram [PRA 98, 052335, Nat Photon 8, 796, PRL 119, 120503], have been proposed to complement fidelity and hence to achieve a more complete assessment of the teleportation systems. Among these criteria, the T-V diagram and the ν - τ diagram evaluate the effective quantum channels simulatable by the teleportation scheme and can be mapped to each other by the following transformations:

$$\nu = \sqrt{V_q}, \text{ and } \tau = T_q \sqrt{V_q} / (2 - T_q).$$

Here T_q stands for the joint signal transfer coefficient defined as

$$T_q = T_x + T_y = \frac{SNR_{xout}}{SNR_{xin}} + \frac{SNR_{yout}}{SNR_{yin}} .$$

It quantifies how much information is successfully recovered. And V_q signifies the additional noise incurred during teleportation and hence dictates how closely the output is correlated to the input. It is defined as

$$V_q = V_{xout|xin} + V_{yout|yin} ,$$

where $V_{xout|xin}$ and $V_{yout|yin}$ represent the input-output conditional variances in the respective quadratures.

In the T - V diagram, an ideal identity channel and hence perfect reconstruction of the input, is embodied by $T_q = 2$ & $V_q = 1$: $T_q = 2$, implying that full knowledge of the input state is obtained by Bob and $V_q = 1$ indicating that the output is maximally correlated to the original input. However, this limit is only achievable with perfect EPR correlation that requires infinitely squeezed sources. In contrast, classical teleportation subject to no entanglement is bounded by $T_q \geq 1$ & $V_q \leq 1$. The constraints are ascribed to the noise penalty imposed onto simultaneous measurement on conjugate quadratures by Heisenberg uncertainty principle. On the other hand, the criteria for a successful quantum teleportation are signatored by $T_q < 1$ & $V_q > 1$, dictating that both the signal transfer coefficient and the input-output correlation have exceeded their corresponding classical limit. In the unity-gain regime, the satisfaction of these criteria is equivalent to the saturation of the no-cloning limit, namely a fidelity $F > 2/3$.

The subfigure (a) below displays the T - V diagram of the conventional deterministic teleportation. To enter the quantum region, both Alice's and Bob's channels are required to have transmission $>50\%$ for any entanglement, and more than -3 dB of initial EPR squeezing. A higher squeezing is required to increase T_q and reduce V_q . The teleporter asymptotically approaches an identity channel when $r \rightarrow \infty$, albeit the maximum attainable T_q equals 2, meaning the total SNR is at best conserved. Therefore, the simulatable channels using conventional setups are confined to the region with $T_q < 2$.

In stark contrast, the heralded quantum teleporter significantly expands the accessible T - V region, as shown in subfigure (b). In a nutshell, the teleporter represents a purified channel akin to that using teleportation and a physical NLA [PRA 84, 022339; PRA 98, 052335; PR Research 2, 013310; Nat Photon 9,764]. If we stick to the unity-gain condition, the protocol empowers one to penetrate into the quantum region by increasing the noiseless gain alone, without demanding a higher initial squeezing. More remarkably, in the gain-tuned regime, the protocol not only mitigates errors incurred during transmission, but also is capable of improving the overall signal-to-noise ratio, giving a signal transfer coefficient $T_q > 2$. This operational region is inaccessible using conventional deterministic setups. The results open up avenues for realizing deterministically impossible tasks, such as quantum telecloning with fidelity surpassing the no-cloning limit [PRA 59, 156] and tele-amplification [Nat Photon 7, 439; PRL 128, 160501]. With a sufficiently high noiseless gain, the heralded teleporter asymptotically approaches a best achievable channel by all means, that is a direct transmission subject to no loss and noise but is embedded by an NLA with noiseless gain of g . In this scenario, the T - V parameters become $T_q = 2g^2$ & $V_q = (1 - g^2)^2$. The results agree with the analysis from Blandino *et al.* in [PRA93, 012326], where they proposed a general framework to describe the effective channel established by a MB-NLA-based teleportation scheme. The transformation enacted by the protocol is equivalent to an effective system comprised of a noiseless amplifier (or attenuator), and a quantum channel which can in principle has no loss and an amount of thermal noise arbitrarily small. The framework is applicable to arbitrary input states including non-Gaussian and non-classical states.

(a) T-V diagram of the deterministic teleporter.

(b) T-V diagram of the heralded teleporter.

This is an interesting topic that certainly merits more investigations. In fact it is the subject of a paper we are preparing. Because the majority of the interesting channels that can be simulated by our protocol are attained under the non-unity gain condition, which strictly speaking, is beyond the scope of the present manuscript where we are mostly concerned with approximating an identity channel. To avoid impairing our future work, we added the following discussions in the main text to address this comment and leave the more detailed discussions reported above to our next paper.

“In this work, we have restricted ourselves to operate under the unity-gain condition, so that our teleporter acts invariantly on all input states. In future, by relaxing this condition, our teleporter can be adapted to simulate a variety of quantum channels otherwise implausible using deterministic setups. The protocol opens up avenues such as quantum tele-amplification and tele-cloning with fidelity surpassing the no-cloning limit, that have been proven useful in coherent-state quantum computing, but are not achievable deterministically. Furthermore, our approach demonstrates an intriguing alternative for mitigating channel loss without physically introducing non-Gaussian operations. As an application, the protocol enables one to distil a Bell state sufficiently to violate the Clauser, Horn, Shimony, Holt (CHSH) inequality.”

And in Sec. III in Methods:

“The results conjure up the proposal by Blandino et al. in [PRA 91, 062305; PRA 93, 012326], where they characterize the effective quantum channel established by a teleporter embedded with a measurement-based NLA. The transformation enacted by the protocol is equivalent to an effective system comprised of a noiseless amplifier (or attenuator), followed by a quantum channel which can in principle have arbitrarily small loss and coupled thermal noise. Our results coincide with this picture. Owing to the channel purification effect, the protocol can be employed to probabilistically purify quantum non-Gaussian states using only Gaussian operations.”

The authors also report a purification effect that their protocol has on input thermal states. My suspicion is that this purification effect is connected with the cut-off in the filtering and with the fact that coherent states with large amplitudes are transferred with less-than-unity gain. A thermal state can be seen as a Gaussian mixture of coherent states. If every coherent state would be teleported perfectly and with unity gain, then also the thermal state would be teleported perfectly. In the considered probabilistic teleportation protocol, the success probability of coherent state teleportation increases with the absolute value of the complex amplitude (up to some cut off). This should in fact lead to increase of thermal noise, since the Gaussian P-function of the thermal state would be multiplied by an inverse Gaussian. However,

if the noise is large enough, then the coherent states with large enough amplitude are transferred with less-than-unity gain due to cut-off in filtering, and the thermal noise is effectively reduced.

Yes, applying our inverse Gaussian filter $f(\alpha_m) \propto e^{-(|\alpha_m|^2 - \alpha_c^2)(1 - 1/g^2)}$ would result in amplified noise. In fact, as a result of the filtering, both the variance and mean amplitude of the input state would be amplified by g^2 . This has been elaborated in [PRA 96, 012319; PRA 93, 012326]. So the reviewer is correct that applying the filter alone does not suffice to purify the displaced thermal input unless significant distortion is incurred due to a small cutoff. This is not what happened in our implementation, where we appropriately simulate a noiseless linear amplifier with a finite but sufficient cutoff. We explain in detail the process in the following.

To perform a measurement-based noiseless linear amplification, one needs to apply a linear rescaling after the post-selection enacted by the inverse Gaussian filter. The local rescaling maps the filtered ensemble α_m to $g\alpha$. The following figure illustrates explicitly the procedure of a measurement-based NLA acting on a Gaussian distribution with mean amplitude α_0 .

Measurement-based NLA. (a) Distribution of the input state, revealed by a direct dual-homodyne measurement. (b) Probability distribution after implementing the filter function with cutoff α_c . (c) The output probability distribution after rescaling which leads to an output with an amplified mean but the same variance as the input.

The reviewer is correct that by restricting the cutoff, the variance of the input can also be reduced; however, the input experiences severe distortions and would lose its Gaussianity as a result. The output variance can be arbitrarily small, but the state is no longer physical, as pointed out in [PRA 96, 012319]. In this work, we always operate the teleporter with sufficient cutoff to ensure a faithful emulation of NLA and avoid over-estimation of the fidelity. This is evidenced in the output probability distribution and the recovered quadrature mean amplitudes presented in Fig. 4.

To clarify this fact, we added the following sentences in the modified main text (citations omitted here for brevity):

“~~ In the measurement-based picture, this post-selection, in conjunction with a local rescaling mapping $\alpha_m \rightarrow g\alpha$, could effectively emulate a noiseless linear amplifier (NLA) with an amplification gain of g . Its operational regime is determined by the cut-off parameter α_c . The approximation to an ideal NLA operation can be made arbitrarily close to perfect by increasing α_c while retaining a finite success probability.”

and in Supplementary Sec. I.

“~ The post-selection, when followed by a rescaling that maps α_m to $g\alpha$, effectively approximates a noiseless linear amplifier.”

It is then questionable what would be the applicability of such purification effect and the authors should discuss in more detail in the manuscript its physical origin to clarify this point.

In response to the reviewer’s comment, we added a new section in Methods in the main text where we provide a way to visualize the basic physics underlying the action of the teleporter on a displaced thermal input state. We derive how the teleporter transforms a displaced thermal input

$$\rho_{in} = D(\alpha_0)\rho_{in}(\lambda_0)D^\dagger(\alpha_0)$$

into another displaced thermal state with mean amplitude and variance of

$$\alpha_{out} = \frac{g\lambda_0'}{1 - g^2\lambda_0'} , \lambda_{out} = g\lambda_0' , V_{out} = \frac{1 + g^2(V_{BS} - 1) + V_{BS}}{1 - g^2(V_{BS} - 1) + V_{BS}} \quad (9)$$

And added the following discussions:

“~ This means for an incident displaced thermal state (ρ_{BS} here), the NLA amplifies its mean amplitude by an effective gain $g_{eff} = \frac{g\lambda_0'}{1 - g^2\lambda_0'} > g$, since $g\lambda_0'$ must remain smaller than 1 for the amplified state to be physical. Consequently, an NLA amplifies a thermal state slightly more than it does to a coherent input state. This fact is essential for realizing the purification effect. For all $1 < g < g_{max} = 1/\lambda_0'$, the signal-to-noise ratio (SNR) of the output is always greater than that of the input state.

To satisfy the unity gain condition, the input and output mean amplitude must be equal, *i.e.* $\alpha_{out} = \alpha_{in}$. As a result of this condition, one obtains the relation between g and r given by

$$g = \frac{\sqrt{V_0^2 - 1 + \tanh(r)^2} - \tanh(r)}{V_0 - 1} \quad (11)$$

Substituting Eq. (11) into Eq. (9), one obtains the overall output variance that has $V_{out} \leq V_{in}$. The rationale behind is that by concatenating an NLA to a beamsplitter (with no excess noise), we can retain the input mean but obtain an output with reduced thermal noise. If we relax the unity-gain condition, by setting $g = g_{max} = 1/\lambda_0'$, the output becomes a pure coherent state.

The results conjure up the proposal by Blandino *et al.* in [PRA 91, 062305; PRA 93, 012326], where they characterize the effective quantum channel established by a teleporter embedded with a measurement-based NLA. The transformation enacted by the protocol is equivalent to an effective system comprised of a noiseless amplifier (or attenuator), followed by a quantum channel which can in principle have arbitrarily small loss and coupled thermal noise. Our results coincide with this picture. Owing to the channel purification effect, the protocol can be employed to probabilistically purify quantum non-Gaussian states using only Gaussian operations.”

Finally, the authors find that in the large gain limit pure input squeezed states can be teleported onto pure states. The authors should clarify whether this requires the knowledge of the quadrature variances of the teleported state and whether the classical gains in the teleportation protocol need to be adjusted according

to these quadrature variances, as implied by Eq. (19). Dependence on the input quadrature variances makes the protocol less general.

We admit that for Gaussian input states other than coherent states, the in-loop rescaling factors $\phi_{x(y)}$ are dependent upon the input quadrature variances in the unity-gain regime. However, we argue that they do not rely on the input amplitudes; therefore, no *a priori* knowledge about the input quadrature amplitudes is required. We added the following sentence in the main text to clarify this fact:

“Our teleporter exhibits an intriguing feature when operating in the regime of large g : given *a priori* knowledge of the quadrature variances of an input Gaussian state, regardless of the initial squeezing level, $V_{xout} V_{yout} = 1$ is approachable for all pure inputs. ~~”

While knowing the input amplitudes could be impractical in some circumstances, it is not impractical to assume that one may have knowledge about the input alphabet. For example, if the input are non-classical and non-Gaussian states such as Fock states, cat states or sub-modes of the EPR states, the dependence on the input variances no longer exist and indeed the best strategy is to operate the teleporter at the gain tuned regime. As shown earlier, our protocol promises improved performance in these circumstances. For input coherent states, our protocol works invariantly on all inputs. For applications such as establishing quantum network [Nat Photon 11, 678; PR Res. 2, 013310; PRA 95, 022312; PRA 102, 052425], quantum key distribution [Nat Comm 6, 8795; Nat 607, 687], distributed quantum computation [Nat 402, 390; PRA 82, 040303; PRL 96, 010503] and sensing [Nat Phys 16, 281; PRL 121, 043604], where resources are prepared offline, our protocol would be beneficial. As another example, the heralded teleporter can be used to simulate quantum channels otherwise implausible using conventional means [PRA 93, 012326; PRA 98, 052335; Quan. Sci. and Tech. 3, 035009; PRL, 119, 120503]. In this regard, one would presumably know the input alphabet. In the discussion, we provided a variety of such situations where the protocol could find useful.

Note also that the pure-state teleportation can be approached if one conditions on measurement outcomes close to zero, i.e. postselects $x_m = y_m = 0$. This has been demonstrated by Akira Furusawa group. It would be worth to compare the present approach with the postselection on x_m and y_m being close to 0, to see what are the trade-offs between the purity and the success probability of the protocol.

This is an illuminating suggestion. We note that the results presented in [PRL 113, 223602] by Furusawa et. al. reported a conditional teleportation that is able to purify a single photon state by adopting a bandpass filter, $x^2 + y^2 \leq L^2$, onto the measurement outcomes of the in-loop dual homodyne. The negativity of the photonic state is resurrected by reducing the threshold parameter L . The filter asymptotically approaches an ideal noiseless linear attenuator (NLAtt) signified by the transformation $|\alpha\rangle \rightarrow \exp[|\alpha|^2/2(1 - g^2)]|g\alpha\rangle$. The exact noiseless attenuation is achieved when $L \rightarrow 0$, whereby the purity of the single photon input is intact. The smaller the cutoff L is, the more faithful the bandpass filter approximates an NLAtt.

We agree that it would be worthwhile comparing this scheme to our protocol regarding trade-offs made between purity (or fidelity) and the success probability.

The protocol in [PRL 113, 223602] essentially provides a measurement-based alternative to the physical implementations of a noiseless attenuator relying on single photon detectors [PRL 109, 180503; AIP Conf. Proc. 1110, 155; Nat. Photonics 9, 764]. The scheme is very efficient in preserving the quantum features of any input state that has a probability distribution or quantum feature such as negativity centered at origin, such as Fock states as demonstrated in the paper, and cat states with small amplitudes. As addressed

above and in Supplementary Sec. II, our scheme is also applicable in this scenario, for purifying the states and hence improving the negativity of the output states. In addition to the above analysis, one may notice that by adopting a gain $g < 1$ in our filter function $f(\alpha_m) \propto \exp[(|\alpha_m|^2 - \alpha_c^2)(1 - 1/g^2)]$ and upon proper normalizations, our filter can be used to effectively emulate a noiseless attenuator instead of a noiseless amplifier. In light of the work by Furusawa *et al.*, we analyze the performance of our filter in this regime acting on a single photon state and a cat state, respectively. We give a brief comparison between the results obtained using Furusawa's bandpass filter and our Gaussian filters. The results are summarized below.

	Our scheme $g < 1$	Inverse Gaussian	Furusawa's scheme
Gain	0.5	1.05	N/A
Truncation	0.25	0.25	0.25
Fock state: fidelity	0.9976	0.9977	0.9976
Fock state: success probability	0.0085	0.0087	0.0087
Cat state: fidelity	0.8732	0.8729	0.8732
Cat state: success probability	0.0034	0.0030	0.0035

Table: Comparing our scheme using a Gaussian filter with $g < 1$ and inverse Gaussian filter with $g > 1$, respectively, and Furusawa's scheme using a bandpass filter. EPR squeezing used is -10 dB in both cases.

It is evident that the two schemes have similar performances in this scenario, where the input states have probability distributions or quantum feature such as negativity centered at origin. However, although the two schemes bear similarities in some circumstances, there exist distinctive differences between the two.

As proved in [PRA 96, 012319], our protocol provides a measurement-based alternative to the physical implementations of an NLA (as well as an NLAtt). The operational regime can in principle be arbitrarily large to encompass any input alphabet, at the expense of a decreasing success probability. In contrast, the bandpass filter would not be applicable for input states containing a high number of photons. Consider a displaced Gaussian input state, the purity of the input can be well preserved so that $\forall x \forall y = 1$ using our protocol, as shown in Eq. (S24). In contrast, using the Furusawa scheme, the high amplitude of the input would demand a high cutoff and hence a degraded purity. This is consistent with the fact that the Furusawa scheme works in analogy to the physical implementations of an NLAtt using photon number detectors. To circumvent the loss or distortion due to a large cutoff, one can build an array of N such devices. In this scenario, the input state is evenly split among N attenuation devices, with each constituent experiencing a noiseless attenuation. The N outputs of the NLAtt network are then combined to reconstruct the input state. The success probability of the noiseless attenuation decreases exponentially with the number of splitting, therefore rendering the protocol very resource intensive in teleporting high-amplitude input states. The scheme, in this scenario, resembles those proposed in [PRL 111, 050504; PR Research 2, 013310].

We added the following paragraphs to briefly compare our schemes to the Furusawa scheme in the Supplementary Sec. II, following the discussions on non-Gaussian input states.

“We note that conditional quantum teleportation relying on a bandpass filter, $x^2 + y^2 \leq L^2$, has demonstrated to be useful for purifying a single-photon input state [PRL113223602]. The negativity of the photonic state is resurrected by reducing the threshold parameter L . A higher purity is obtained by decreasing L , at the expense of a reduced success probability. The underpinning idea is to asymptotically approach an ideal noiseless linear attenuator (NLAtt) signified by the transformation $|\alpha\rangle \rightarrow \exp[|\alpha|^2/2(1 - g^2)]|g\alpha\rangle$. The exact noiseless attenuation is achieved when $L \rightarrow 0$, whereby the

purity of the single photon state stays intact. One may notice that when operating our scheme with a noiseless gain $g < 1$ in Eq. (9), upon proper normalizations, our protocol effectively emulates a noiseless linear attenuator instead of a noiseless amplifier. In light of this discovery, we adopt a noiseless gain $g < 1$ for post-selection and analyze the action of our teleporter in this operation region on the same single photon and cat input states used above. We obtain output fidelity and success probability of $F = 0.9976$, $P_s = 0.0085$ for the single photon input and $F = 0.8732$, $P_s = 0.0034$ for the cat input state, respectively. The protocol works comparably to the scheme in [PRL113223602], whereby the corresponding fidelity and success probability are $F = 0.9976$, $P_s = 0.0087$ and $F = 0.8732$, $P_s = 0.0035$, respectively. This is to be expected – the two protocols have similar performance in circumstances where the input states have probability distributions or quantum feature such as negativity centered at the origin.

However, note that the protocol in [PRL113223602] essentially provides a measurement-based alternative to the physical implementations of a noiseless attenuator relying on single photon detectors [PRL 109, 180503; AIP Conf. Proc. 1110, 155; Nat. Photonics 9, 764]. Its operating region is consequently restricted to states with small amplitudes. In contrast, the operation regime of our protocol can in principle be arbitrarily large to encompass any input alphabet, at the expense of a decreasing success probability. To apply the protocol in [PRL113223602] for input states containing a high number of photons, one can build an array of N such devices. The input state is evenly split among these N setups, with each constituent experiencing a noiseless attenuation. The N outputs of the NLAtt network are then combined to reconstruct the input state. The price to pay is that the success probability would decrease exponentially with the number of splitting. The scheme, in this scenario, resembles those proposed in [PRL 111, 050504; PR Research 2, 013310].”

We thank the reviewer for his/her support and the constructive comments. The reviewer's highlighted various instances where further analysis and discussions could be added. We have addressed all the points raised, and applied significant changes upon our manuscript. We added two new Supplementary sections and a new section in Method in the main text. Various other changes have been made accordingly. We hope that the reviewer is satisfied with our response.

Reviewer #2

We thank the reviewer for the detailed reading and helpful comments. Although the reviewer appreciates the improved fidelity and reduced thermal noise addition on teleportation of coherent states, the reviewer raised a concern about the applicability of the setup onto non-classical and non-Gaussian input states. The comments raised an interesting question. We address this comment below.

The submitted manuscript presents an experimental realization of a teleportation protocol that utilizes noiseless amplification to improve the fidelity of the transmitted coherent states at the cost of success rate. The amplification is based on probabilistic post-selection of the results of the bell-type measurement. The obtained experimental results demonstrate that coherent states can be indeed teleported with a reduced amount of added thermal noise and improved fidelity.

The article then claims to demonstrate a 'near unit fidelity when transmitting quantum states over long distance channels, however significant the channel loss is.' However, this is demonstrated only by using coherent states, which are both Gaussian and classical, and solely described by their first moments of quadrature operators. As a consequence, there are protocols that work on coherent states, such as those in references [41, 42], that are not suitable for general quantum states required for quantum computing tasks. The theoretical discussion of the protocol is also at the level of moments of quadrature operators and therefore also valid for coherent (or Gaussian states). It is not clear whether the unity gain regime works for different quantum states. It is also not clear, whether the purification effect is obtained for other than coherent states, or whether it works for all coherent states or just a subset. This also means that the claim in the conclusion about using the proposed amplifier as a quantum repeater is not supported by the rest of the manuscript.

In summary, my main issue is that the manuscript does not discuss the limitations of the setup. The performance of the setup is demonstrated on several coherent states but this fact and its consequences are glossed over in the discussion. If the protocol can be used also for non-classical and non-Gaussian quantum states, discussing and demonstrating this would make the article suitable for publication in Nature communication. On the other hand, if the protocol only works for coherent states, I would recommend a different and more specialized journal.

We agree with the reviewer that it is important to address the limitations of the setup and its applicability onto non-classical and non-Gaussian states. We also appreciate that the reviewer thinks if the protocol can be used also for these states, discussing and demonstrating this would make the article suitable for publication in Nature Communication.

- 1) First, in response, we extend our simulations based on the moments of quadrature operators. We provide a general formalism that describes the working mechanism of the heralded quantum teleporter using Wigner functions. We perform some preliminary investigations and show that the protocol can be extended to non-classical and non-Gaussian input states. We added a new section (Sec. II) in the Supplementary Material to address the results.

We report explicit results as two prime examples where we consider a single photon input Fock state and an input cat state because these non-classical states are crucial in many quantum information protocols. We find that the filter function can be optimized so that the protocol is applicable to non-classical and non-Gaussian quantum states apart from the coherent states considered originally – higher fidelity is achievable without requiring more squeezing resources.

Figure S5 shows the output Wigner functions of the heralded teleporter for a single photon input (a) and a cat input (b), given -10 dB initial squeezing resource. In comparison with a deterministic teleporter, significant enhancement in the output fidelity is attainable in both circumstances with a practical success probability. In Fig. S6, we plot the output Wigner functions at $x=0$ to show implicitly how the negativity and quantum coherence of the input states are well preserved using the current teleportation scheme. The capability to preserve the quantumness of input states constitute another measure of the teleporter's performance. This measure complement fidelity and is very suitable in the context of non-Gaussian input states [Nat 566 509, Nat Nanotech 12 1026, Science 370, 1460, Nat Photon 5 52]. As shown in Fig. S6, our teleporter shows improved performance in retaining the negativity of the input states with a moderate level of initial squeezing. In particular, the negativity of the single photon state is almost intact which would require infinite squeezing using conventional deterministic setups.

Fig. S6. Wigner functions $W(0,y)$ of the input and output states from our heralded quantum teleporter and the deterministic teleporter, when benchmarked on the same entangled resource.

Fig. S5. \mathcal{F} and P_s are the output fidelity and success probability of the heralded quantum teleporter, respectively, whereas \mathcal{F}_{DET} is the obtainable fidelity using deterministic teleporters.

In addition, we provide analysis from a different perspective using different means. In light of the work reported in [PRL 113, 223602; PRL 109, 180503; Nat Photonics 9, 764], we show that our scheme with $g < 1$ in the filter function $f(\alpha_m) \propto \exp[(|\alpha_m|^2 - \alpha_c^2)(1 - 1/g^2)]$ effectively approximates a noiseless linear attenuator which was proven to be useful in preserving the purity of these non-Gaussian and non-Classical states. We obtained fidelities and success probabilities of $F = 0.9976, P_s = 0.0085$ for an input single photon state and $F = 0.8732, P_s = 0.0034$ for a cat state. These results are similar to the ones obtained using the inverse Gaussian filter, as shown in Fig. S5. However, the impact of these investigations lies in illuminating the more general applicability of our scheme, whereby its operational regime should not be restricted to $g > 1$, but instead to be appropriately tailored relative to the input alphabet.

- 2) Second, we added a new section in the Supplementary Material (Sec. V) to show that the heralded quantum teleporter can be used for teleporting one sub-mode of an EPR state and results in improved quantum correlations successfully shared between Alice and Bob. The scheme constitutes an advanced entanglement swapping protocol, which is an essential building block in constructing quantum repeaters. The performance of the protocol is evaluated by two entanglement witnesses that haven't been used previously in deterministic swapping protocols: first, an inequality adopted by Tan *et al.* in the deterministic entanglement swapping protocol [PRA 60, 2752] $\langle \Delta(X_A - X_B)^2 \rangle + \langle \Delta(Y_A + Y_B)^2 \rangle < 4$, a sufficient condition for entanglement and second, the *entanglement inseparability* proposed by Duan *et al.* [PRL 60, 2752], a necessary and sufficient condition for entanglement.

We find that owing to the NLA, the inequality can be saturated with less initial EPR squeezing, as compared to a deterministic scheme [PRA 61, 010302, PRA 60, 2752, PRA 68, 012319]. Interestingly, there exists an operational regime where the conditional variance equals $4e^{-2r_1}$ where r_1 is the squeezing parameter of the EPR state to be teleported. As such, the input EPR state's quantumness is fully preserved. Note that using our protocol, this is achievable with moderate entangled resources, which would otherwise require infinite squeezing using the deterministic swapping schemes.

The heralded teleporter also results in improved *entanglement inseparability* as compared to the conventional deterministic schemes regardless of operating whether in the unity-gain regime or in the more general gain-tuned regime.

Fig.S10. Entanglement inseparability of the EPR sub-modes successfully shared between Alice and Bob. (a) operating the heralded teleporter under non-unity gain condition, (b) under unity-gain condition. r_2 is the squeezing parameter of the EPR channel in teleportation.

In the circumstances discussed above, the noiseless gains are not optimized. It would be interesting to work out an explicit expression of the optimal noiseless gain and the limitation on the gain values. It is also worthwhile adopting other entanglement measures to assess the protocol. We leave these subjects to future investigations.

The major compromise made in our protocol is the trade-off between success probability and the mitigation of decoherence. We stress that our protocol is heralded by adding the following sentence in the discussion.

“The enhancement in mitigating decoherence is made possible by trading determinism.”

Albeit being probabilistic, there are a number of situations where trading determinism for an enhanced teleportation efficacy can be beneficial. We added the following paragraph to highlight these scenarios. We repeat our response here (citations omitted here for brevity).

“Our work opens up a number of interesting future directions. High-fidelity quantum teleportation, in its own right, has distinct merits in a broad class of quantum information and computation applications. Notably, in the Supplementary Sec. V, we show that our heralded teleporter when applied to EPR entanglement results in improved quantum correlations successfully shared between the end users. In this regard, the teleporter works as an entanglement swapping protocol that outperforms conventional deterministic swapping schemes. This provides a feasible pathway to reducing the resource overhead in constructing regenerative relays and hence quantum repeaters when highly squeezed quantum resources are not available. In distributed quantum computing and sensing, and quantum key distribution where resources are prepared offline, trading determinism for higher fidelity and enhanced entanglement distribution can be beneficial. In this work, we have restricted ourselves to operate under the unity-gain condition, so that our teleporter acts invariantly on all input states. In future, by relaxing this condition, our teleporter can be adopted to simulate a variety of quantum channels otherwise implausible to access using deterministic setups. The protocol opens up avenues such as quantum tele-amplification and tele-cloning with fidelity surpassing the no-cloning limit, that have been proven useful in coherent-state quantum computing, but are inconceivable deterministically. Furthermore, our approach demonstrates an intriguing alternative for mitigating channel loss without physically introducing non-Gaussian operations. As an application, the protocol enables one to distill a Bell state sufficiently to violate the Clauser, Horn, Shimony, Holt (CHSH) inequality. Each of the above possibilities merit investigations and could be of interest from either practical or fundamental perspectives.”

We thank the reviewer for his/ her helpful and constructive comments. We have addressed all the points raised and included new results for non-classical and non-Gaussian input states. We hope that the reviewer is satisfied with our response.

Reviewer #3

We appreciate that the reviewer thinks our results will surely be of interest to many in the community, especially with record-high teleportation fidelities and the flexibility in its operation. We thank the reviewer for the detailed reading and instructive comments. We have made changes accordingly to the manuscript to improve the clarity of presentation. We address the reviewer's comments below.

Jie Zhao and colleagues present an interesting spin on the traditional continuous-variable teleportation protocol that can attain higher than usual fidelities at the expense of probabilistic operation. In fact, it can approach unit fidelity, even for realistic squeezing levels. The scheme is demonstrated experimentally for teleportation of coherent states and thermal states, and also includes tele-amplification where the output coherent state amplitudes are boosted.

The protocol makes use of a variant of the virtual noiseless linear amplifier (NLA) that was developed by some of the authors (and others) and implemented by them for several different applications. The virtual NLA works by filtering or post-selecting measurement outcomes and is much more experimentally feasible than a physical NLA. The experimental scheme employed here is essentially the good old CV teleporter constructed from two-mode squeezed light, a Bell (dual homodyne) measurement, and active feed-forward. However, now the teleported states are only kept when the intermediate Bell measurement outcomes pass a certain filtering test which accepts high-amplitude outcomes with higher probability than outcomes close to the origin of phase space. This filtering effectively biases the statistics of the post-selected output states towards larger amplitudes, hence acting as an amplifier. Since this amplification is noiseless (made possible by the probabilistic/heralded operation) and takes over from part of the feed-forward gain applied in the standard teleportation protocol, the teleported states suffer less from the noise added due to finite squeezing of the entangled resource state. Interestingly, the protocol can also act as a purifier of input thermal states, basically through the attenuation + noiseless amplification process.

Since the protocol is heralded - in fact, success probabilities are rather low - it will have reduced applicability compared to standard CV teleportation which is famously deterministic. Still, it will surely be of interest to many in the community, especially with the record-high teleportation fidelities and the flexibility in its operation. The experimental results are convincing, and the paper contains a good discussion of the protocol's relation to similar purification schemes previously proposed or demonstrated.

We are glad that the reviewer finds the protocol interesting and the experimental results convincing.

However, the presentation is let down by somewhat sloppy writing. Most glaringly, the working principles of the protocol and the actual experimental implementation are very poorly explained. It was essentially impossible for me to understand this without studying the supplement and some of the group's earlier papers - even though I was roughly familiar with those prior works. While other readers will certainly be brighter than me, I am sure I will not be the only one left baffled by the central section explaining the protocol - essentially page 2, left column. Here are some of the problems with this passage:

We agree with the reviewer that it is essential to the appreciation of the paper that the protocol is properly explained. And we are sorry that the reviewer felt not sufficient details were provided. We have restructured and re-written a significant portion of page 2, left column and have added two new paragraphs accordingly to include more details on the working principles of the protocol and the experimental implementations. We repeat our response here (citations omitted here for brevity).

“The experimental schematic of our teleporter is depicted in Fig. 1a. Two squeezed single modes with squeezing parameter r are combined to create a two-mode squeezed state, also referred to as the Einstein-Podolsky-Rosen (EPR) state. One arm of the EPR state is coupled to an input state ρ_{in} on a 50:50 beamsplitter and detected on a dual-homodyne station simultaneously measuring the amplitude and phase quadratures. The measurement outcomes, denoted as $\alpha_m = (x_m + iy_m)/2$, undergo a filtering algorithm embodied by an acceptance function given by

$$f(\alpha_m) \propto \exp[(|\alpha_m|^2 - \alpha_c^2)(1 - 1/g^2)]$$

if the quadrature amplitude $|\alpha_m| < \alpha_c$; otherwise, the measurement ensemble is kept with unity probability. In the measurement-based picture, this post-selection, in conjunction with a local rescaling mapping $\alpha_m \rightarrow g\alpha$, could effectively emulate a noiseless linear amplifier (NLA) with an amplification gain of g . Its operational regime is determined by the cut-off parameter α_c . The approximation to an ideal NLA can be made arbitrarily close to perfect by increasing α_c while retaining a finite success probability. When successful, the heralded amplitude and phase records are rescaled electronically by ϕ_x and ϕ_y respectively, and broadcast to Bob via a classical communication channel. Bob performs a displacement operation on his EPR mode accordingly to reconstruct the input state.

The working mechanism of the teleporter is conceptually illustrated in Fig. 1b. We first recall the conventional quantum teleportation scheme. In this diagram, the teleportation can be decomposed into a pre-attenuation followed by a post-amplification. The in-loop classical rescaling factor ϕ is first tuned to $\beta = \lambda\sqrt{2}$ where λ denotes the entanglement parameter $\lambda = \tanh(r)$. Conventionally, this corresponds to the *gain tuning* operational point where the teleporter passively attenuates the input without injecting excess noise. A phase-insensitive linear amplification with gain $\epsilon = 1/\lambda$ is then exploited to satisfy the unity-gain condition, whereby the output has the same quadrature amplitudes as the input. Unity-gain justifies the universality of a quantum teleporter which acts invariantly upon arbitrary input states. As elucidated by Haus and Mullen and Caves, any phase-insensitive linear amplification is unavoidably subject to a noise penalty equivalent to $|\epsilon^2 - 1|$ units of vacuum noise. The teleported state, therefore, has increased quadrature variances as compared to the input state. It is only possible to avoid this additional noise and hence achieve unity fidelity when $\epsilon \rightarrow 1$, that is $r \rightarrow \infty$. In contrast, the heralded teleporter incorporates a noiseless linear amplifier to complement the deterministic amplification. The interplay between the two distinctive amplification schemes fulfills the unity-gain condition. The noiseless gain g and the deterministic gain $\epsilon < 1/\lambda$ are tuned with a high premium given to optimizing the output fidelity at a practical success probability. With an increased g , thereby decreased ϵ to conform to unity gain, one can achieve larger noise reduction, and hence higher fidelity enhancement surpassing the quantum-limited performance for a given entanglement resource. ~~”

1. The authors seem to assume that the reader is intricately familiar with previous works on virtual noiseless amplification (like they themselves of course are).

We agree that the virtual noiseless amplification, which empowers the present heralded teleportation scheme, should be properly elaborated, especially for the uninitiated readers. We adopt the reviewer’s suggestion and have added the explicit form of the post-selection acceptance function along with some descriptions.

“The measurement outcomes, denoted as $\alpha_m = (x_m + iy_m)/2$, undergo a filtering algorithm embodied by an acceptance function given by

$$f(\alpha_m) \propto \exp[(|\alpha_m|^2 - \alpha_c^2)(1 - 1/g^2)]$$

if the quadrature amplitude $|\alpha_m| < \alpha_c$; otherwise, the measurement ensemble is kept with unity probability. In the measurement-based picture, this post-selection, in conjunction with a local rescaling mapping $\alpha_m \rightarrow g\alpha$, could effectively emulate a noiseless linear amplifier (NLA) with an amplification gain of g . Its operational regime is determined by the cut-off parameter α_c . The approximation to an ideal NLA can be made arbitrarily close to perfect by increasing α_c while retaining a finite success probability.”

2. There is essentially zero explanation of the experimental scheme (Fig. 1a). All discussion is concerned with the dynamics illustrated in Fig. 1b. The description of the experiment is left for the figure caption and the methods section.

This is a useful comment. We agree that the experimental setup was not properly introduced in our original manuscript. We have added descriptions in this regard.

“The experimental schematic of our teleporter is depicted in Fig. 1a. Two squeezed single modes with squeezing parameter r are combined to create a two-mode squeezed state, also referred to as the Einstein-Podolsky-Rosen (EPR) state. One arm of the EPR state is coupled to an input state ρ_{in} on a 50:50 beamsplitter and detected on a dual-homodyne station simultaneously measuring the amplitude and phase quadratures. The measurement outcomes, denoted as $\alpha_m = (x_m + iy_m)/2$, undergo a filtering algorithm embodied by an acceptance function given by $\sim\sim$ if the quadrature amplitude $|\alpha_m| < \alpha_c$; otherwise, the measurement ensemble is kept with unity probability. $\sim\sim$ When successful, the heralded amplitude and phase records are then rescaled electronically by ϕ_x and ϕ_y respectively, and broadcast to Bob via a classical communication channel. Bob performs a displacement operation on his EPR mode accordingly to reconstruct the input state. $\sim\sim$ ”

3. Because of this, the text "where λ denotes the entanglement parameter [...] and r being the squeezing parameter" is hanging in the air, referring to nothing previously mentioned.

We have unambiguously defined the parameters and explained how they are linked to each other.

4. Similarly, there is no mention of the central element of the scheme, namely the post-processing filter. Due to this (and probably other missing explanations), it does not become clear how the noiseless amplification actually is performed. I also found that it would be useful to have a bit more detail in the NLA processing panel in Fig. 1 - perhaps including details from Fig. 5 in Methods and Fig. 1 in the appendix.

We have amended the relevant paragraphs to include the post-selection acceptance function in the main text. We have also added a scheme of the post-selection similar to that in Fig.5 into Fig.1 to show explicitly how the filtering is implemented.

5. It is unclear what is meant by "an arbitrary unknown input state is attenuated with an electronic rescaling factor". This refers to a scaling of the measurement outcomes, but sounds like actual attenuation of the input state, especially since nothing has yet been said about the data processing.

The reviewer is correct that the rescaling refers to a scaling of the measurement outcomes rather than a physical attenuation of the input state. We have re-written the whole section to explain the working mechanism of the teleporter in a more precise and detailed way.

6. The sentence beginning "By virtue of ϵ being arbitrarily close to unity" does not make sense to me, even after re-reading the paper and the sentence several times.

We thank the reviewer for pointing out this ambiguity. We have deleted this sentence.

7. It is unclear whether the scheme works for truly arbitrary input states, only for Gaussian states, or only for coherent states. The paragraph starts with "arbitrary unknown input state" but ends with an expression for the output variances, which I assume is only valid for the Gaussian states that are treated in the supplement.

This is a valid concern which was also raised by the other reviewers. In response, we extend our simulations that are based on the moments of quadrature operators to a general framework using Wigner functions.

We provide results on the performance of our teleporter acting on non-classical and non-Gaussian input states, such as a single photon state and a cat state. A new section (Sec. II) was added in the Supplementary material in this regard. In both circumstances, the heralded teleporter exhibits improved fidelity and enhanced preservation of the input states' negativity and quantum coherence.

8. The relation between the many different parameters (ϵ , g , β , $\varphi_{x,y}$) and their meaning is somewhat opaque, especially since there are basically no formulas in the main text.

The reviewer is correct: the parameters were not defined precisely in the previous manuscript. By re-writing page 2, left column, we provide explicit definitions of the parameters and their relationships.

9. "...complete removal of excess noise is obtained at the expense of a vanishing success probability" - this is the first time that it is explicitly mentioned that the protocol is probabilistic (apart from the word "heralded"). It is not explained why it is so.

In the revised manuscript, we stress that our protocol is heralded by adding the following sentence in the discussion.

"The enhancement in mitigating decoherence is made possible by trading determinism."

In addition, we show explicitly the post-selection filter function in the main text and address how measurement outcomes of the dual-homodyne are selected, as addressed above.

I encourage the authors to rewrite this section, putting on the glasses of someone who understands standard CV teleportation but is perhaps unfamiliar with NLAs and in particular the virtual kind. If length becomes an issue, some of the introduction or discussion could perhaps be sacrificed, or the figure size or caption text be reduced. I find it essential to the appreciation of this paper that the protocol is properly explained. It might also be useful to refer to their earlier work (or other references) at appropriate points in this section.

We thank the reviewer for the instructive comments and suggestions. We agree that it is essential to the appreciation of the paper that the protocol is properly explained. As stated above, we have restructured and made major revisions on the relevant parts of the manuscript. We have added paragraphs to describe the post-selection, how it is used to emulate an NLA and how to interpret the inclusion of NLA as complement to the deterministic amplification in the conventional teleportation schemes, which are

essential for achieving significant enhancement in fidelity. We believe the clarity of presentation has been significantly improved.

A few other points for consideration:

* A brief discussion of possible scenarios where the scheme would be beneficial in spite of its probabilistic nature would be instructive.

We thank the reviewer for this constructive suggestion. We have added a new paragraph to highlight several scenarios where our protocol is useful in spite of its probabilistic nature. In particular, there are situations where trading determinism empowers one to achieve many tasks otherwise implausible using conventional means. We repeat our response here (citations omitted here for brevity).

“Our work opens up a number of interesting future directions. High-fidelity quantum teleportation, in its own right, has distinct merits in a broad class of quantum information and computation applications. Notably, in the Supplementary Sec. V, we show that our heralded teleporter when applied to EPR entanglement results in improved quantum correlations successfully shared between the end users. In this regard, the teleporter works as an entanglement swapping protocol that outperforms conventional deterministic swapping schemes. This provides a feasible pathway to reducing the resource overhead in constructing regenerative relays and hence quantum repeaters when highly squeezed quantum resources are not available. In distributed quantum computing and sensing, and quantum key distribution where resources are prepared offline, trading determinism for higher fidelity and enhanced entanglement distribution can be beneficial. In this work, we have restricted ourselves to operate under the unity-gain condition, so that our teleporter acts invariantly on all input states. In future, by relaxing this condition, our teleporter can be adopted to simulate a variety of quantum channels otherwise implausible to access using deterministic setups. The protocol opens up avenues such as quantum tele-amplification and tele-cloning with fidelity surpassing the no-cloning limit, that have been proven useful in coherent-state quantum computing, but are inconceivable deterministically. Furthermore, our approach demonstrates an intriguing alternative for mitigating channel loss without physically introducing non-Gaussian operations. As an application, the protocol enables one to distil a Bell state sufficiently to violate the Clauser, Horn, Shimony, Holt (CHSH) inequality. Each of the above possibilities merit investigations and could be of interest from either practical or fundamental perspectives.”

* The short paragraph describing the teleamplification result appears abruptly without much introduction or connection to the previous text.

We thank the reviewer for pointing this out. We have deleted the original short paragraph about tele-amplification and instead have included the tele-amplification in discussion. We repeat the relevant part below (citations omitted here for brevity).

“In this work, we have restricted ourselves to operate under the unity-gain condition, so that our teleporter acts invariantly on all input states. In future, by relaxing this condition, our teleporter can be adopted to simulate a variety of quantum channels otherwise implausible to access using deterministic setups. The protocol opens up avenues such as quantum tele-amplification and tele-cloning with fidelity surpassing the no-cloning limit, that have been proven useful in coherent-state quantum computing, but are inconceivable deterministically.”

* Section 1 of the appendix could benefit from some polishing:

At the reviewer's suggestion, we have polished and restructured the first section for easier reading. We have expanded some of the paragraphs to include more detailed explanations of some equations. In addition, we have fixed all the typos and have proof-read the section a few times to avoid any mistakes.

* Some steps are not explained well, e.g. 2nd equality of eq. (6),

We thank the reviewer for pointing out the ambiguities. We have added more explanations to help clarify the derivations.

More specifically, we added the following sentences after Eq. (6) to explain the meaning of the conditional variances:

"The conditional variance $\Sigma_{X_{\{2\}}|\bar{x}_{\{3\}}}$ refers to the amplitude quadrature variance of the mode X_2 conditioned upon the results obtainable in the quadrature measurements of mode X_3 . It quantifies how well one can estimate the amplitude quadrature of mode 2 based on observed amplitude values of mode 3 up to, on average, some noise $\langle X_{\{3\}} X_{\{3\}} \rangle$. For example, given an EPR state with its two constituents being modes a and b , the conditional variance would be [Rev. Mod. Phys. 77 513 (2005)]

$$\Sigma_{X_{\{a\}}|\bar{x}_{\{b\}}} = \langle X_{\{a\}} X_{\{a\}} \rangle \left(1 - \frac{\langle X_{\{a\}} X_{\{b\}} \rangle^2}{\langle X_{\{a\}} X_{\{a\}} \rangle \langle X_{\{b\}} X_{\{b\}} \rangle} \right) = \frac{1}{\cosh(2r)}.$$

Estimation of the quadrature amplitudes of one mode upon the measurement outcomes of the other is subject to the maximum amount of noise when $r \rightarrow 0$; otherwise, perfect EPR correlation is attainable when $r \rightarrow \infty$, leading to $\Sigma \rightarrow 0$. Perfect estimation is therefore achievable."

eq. (10),

Equation (10) (Eq. (13) in the current manuscript) shows the action of the filtering on the classical ensemble α_m , that is to amplify the mean and variance of α_m both by g^2 . We have added explanations on the working mechanism of the post-selection after Eq. (8) (Eq. (9) in the current manuscript) and have amended the description of Eq. (10) (Eq. (13) in the current manuscript) as follows:

"In practice, all measurement outcomes with magnitude less than α_c , namely $|\alpha_m| < \alpha_c$, are selected with probability specified in Eq. (9); outcomes that fall beyond the cut-off circle are kept with unity probability. ~"

To be concrete, assume the dual-homodyne records measurement outcomes α_m that follow an unnormalized Gaussian distribution with mean α_0

$$p(\alpha_m) \propto e^{-|\alpha_m - \alpha_0|^2}.$$

The output distribution after post-selection becomes proportional to

$$p(\alpha_m)f(\alpha_m) \propto e^{-|\alpha_m - \alpha_0|^2} e^{(|\alpha| - \alpha_c^2)(1 - 1/g^2)} = \frac{N e^{(g^2 - 1)|\alpha_m|^2}}{e^{\alpha_c^2(1 - 1/g^2)}} e^{-|\alpha - g^2 \alpha_m|^2/g^2}.$$

Overall, the post-selection gives rise to a displaced distribution with the input mean and variance both being amplified by g^2 . Although this transformation is demonstrated for Gaussian input ensembles, it

holds valid for other states due to its universality in emulating a physical noiseless linear amplifier [PRA 96 012319]. Upon post-selecting the dual-homodyne outcomes, the corresponding mode quadrature operators effectively undergo the following transformations

~~

We emphasize that the transformation embodied in Eq. (12) does not represent a physical unitary process, but rather change in the classical statistical properties of the modes' amplitude and phase quadratures.

eq. (16).

We modified Eq. (16) (Eq. (19) in the current manuscript) and added the following descriptions:

“The post-selected signals $\tilde{X}_{\{3\}}$ and $\tilde{Y}_{\{1\}}$ are rescaled electronically by $\phi_{x(y)}$ prior to being sent to Bob through a classical channel. Bob performs a displacement operation onto his EPR mode using a pair of electro-optical modulators. Therefore, the mean quadrature amplitudes of the teleported state can be expressed as mean of the EPR mode 2 plus a displacement ~”

* In relation to eq. (2), a description of the initial squeezed input states would be in place.

We agree that a description of the initial squeezed input states could be helpful. We have added the following sentences after Eq. (2).

“Here $r_{\{Ax(y)\}}$ and $r_{\{Bx(y)\}}$ refer to the squeezing parameters of the initial single-mode input states A and B, respectively. The subscript $x(y)$ denotes the amplitude (phase) quadrature. Note that the diagonal elements $C_{\{11\}}$ and $C_{\{22\}}$ represent the thermal noise variance in the quadratures of the EPR state, while the off-diagonal elements $C_{\{13\}}$ and $C_{\{24\}}$ are the EPR correlations between the quadratures.”

* The notation for correlations is all over the place, using Δ , δ , implied Δ , and V in one big mixture.

We apologize for this inconsistency in expression. We have deleted δ and V in the derivations and used only Δ to denote the correlations throughout the text.

* Is it possible to write an explicit expression for $\phi_{x(y)}$?

We have provided the more explicit expressions for $\phi_{x(y)}$ in Eq. (22).

* $\langle \rangle$ missing around $Y_1 Y_1$ in (7).

We have added $\langle \rangle$ around $Y_1 Y_1$ in Eq. (7) (Eq. (8) in the current manuscript).

* The tilde-correlation appears to be on the wrong side of the fraction in the Y -expression, first line of eq. (14).

We thank the reviewer for pointing out the typo. We have moved $\langle \tilde{Y}_{\{1\}} \tilde{Y}_{\{1\}} \rangle$ from the denominator to the numerator.

* Prime that shouldn't be there in (13).

We have deleted the prime on $\tilde{X}_{\{3\}}$.

* The mode order used (stated in (5)) is not so helpful.

We thank the reviewer for pointing out this. We have added some sentences before and after Eq. (5) to clarify the definition of the covariance matrix, the quadrature field operators and their relations.

“~~~ where the subscripts 1,2,3 denote the respective spatial modes as illustrated in Fig.1. Elements of the covariance matrix \mathbf{C}' in Eq. (4) are defined accordingly as $\mathbf{C}'_{\{ij\}} := 1/2 \langle \{\Delta X_{\{i\}}, \Delta X_{\{j\}}\} \rangle$, where $\Delta X_{\{i\}} := X_{\{i\}} - \langle X_{\{j\}} \rangle$ ” and $\{,\}$ is the anticommutator. As an example, $C_{\{11\}} = \mathbf{C}'_{\{11\}}$ equals to $= 1/2 \langle \{\Delta X_{\{2\}}, \Delta X_{\{2\}}\} \rangle$, that is the noise variance in amplitude of mode 2. Therefore, the covariance between the amplitude quadratures of modes n and m can be written as $\langle \Delta X_{\{n\}} \Delta X_{\{m\}} \rangle$. For brevity, Δ is omitted in the following derivations. ~~~”

We thank the reviewer for the detailed report and the many helpful suggestions. We have addressed all the points raised which helped improve the presentation and clarity of the manuscript. We hope that the reviewer is satisfied with our response.

REVIEWER COMMENTS

Reviewer #1 (Remarks to the Author):

The authors have provided a comprehensive reply to my previous comments and have made significant changes and amendments to the manuscript. However, the added material triggers some additional questions and raises doubts about the validity of some parts of the analysis and conclusions presented by the authors. In what follows, I will focus on the discussion of entanglement swapping presented in Section V of Supplementary material.

The authors claim that using the heralded teleporter they can outperform standard deterministic CV entanglement swapping schemes. The considered protocol is illustrated in Fig. 8 of Supplementary Material. Let us first consider a deterministic entanglement swapping. Assume that the two shared entangled states are two-mode squeezed vacuum states with squeezing constants r_1 and r_2 , respectively. The input state is a pure Gaussian state and the two-mode homodyne detection in the CV entanglement swapping is a Gaussian measurement. Therefore, for each particular measurement outcome, the state of Alice and Bob will be a pure Gaussian state, in particular a coherently displaced two-mode squeezed state. Note that the covariance matrix of the conditional state is fixed and does not depend on the measurement outcome, only the coherent displacement is a function of the measurement outcome. Alice and Bob can undo this displacement to deterministically recover a two-mode squeezed vacuum state with squeezing constant $\tanh(r) = \tanh(r_1)\tanh(r_2)$. Note that the optimal corrective displacement generally need not be unity-gain and the feedforward rescaling factors should be optimized depending on r_1 and r_2 .

Let us now switch to the probabilistic heralded protocol. This protocol will filter the ensemble of conditional states, i.e. it will modify the probabilities of each conditional state (that corresponds to a specific outcome of homodyne detection). However, all these conditional states have the same covariance matrix M and they differ only by coherent displacements. Therefore, covariance matrix G of any statistical mixture of such states will satisfy $G \geq M$, hence the post-selection and filtering cannot improve the entanglement. There is also another argument why the filtering proposed by the authors should not work,

and this is based on the no-go theorems for entanglement distillation of Gaussian states with Gaussian operations. Let $r_1 > r_2$. It appears that the authors claim that their scheme can achieve entanglement swapping where the output squeezing is equal to r_1 , i.e. $r_{out} = r_1$. Suppose now that Alice and Bob share the two-mode squeezed vacuum with parameter r_2 . Alice locally prepares the two-mode squeezed vacuum state with parameter r_1 and then Alice and Bob apply the heralded protocol. Using local Gaussian operations, classical communication and conditioning they would probabilistically increase the entanglement shared between Alice and Bob, which is known to be impossible.

The above discussion suggests that the analysis presented by the authors may not be completely reliable and therefore the manuscript is not suitable for publication.

Reviewer #2 (Remarks to the Author):

I would like to thank the authors for revising and expanding the manuscript. Showing that the technique can, in principle, work even for non-Gaussian and entangled states significantly contributes to its usability and possible impact.

I believe the technique can be seen as a bridge between the deterministic teleportation based on a feed-forward and probabilistic teleportation based on post-selection with the added benefit of not enforcing the attenuation that the probabilistic teleportation requires. In this sense, the presented method naturally expands the existing toolbox of quantum optics experiments and will most likely see further use or at least testing.

I therefore recommend the manuscript for publication.

If I may have one final recommendation, I would suggest that, in the places of the manuscript where the new technique is compared to the deterministic teleportation, authors also compare it to the the probabilistic teleportation based on post-selecting on a single value of homodyne measurement. For the some scenarios, such as teleportation of a photon or entanglement swapping, this approach provides significantly better results than deterministic teleportation and it would benefit the reader considerably to see a full and fair

comparison with the state-of-the-art.

Reviewer #3 (Remarks to the Author):

The authors made an impressive effort to address all the reviewers' comments and went beyond what would usually be expected in terms of clarifications and new material in the revised version. They clearly addressed all my comments. The paper is significantly improved with respect to readability and impact, and I can now surely recommend publication in NatComm.

Reviewer #1

We thank the reviewer for the time reviewing our manuscript. We appreciate the reviewer's thorough examination of the new supplementary materials that was added in response to the reviewer's comments. We agree with some concerns raised by the reviewer and realized that some of our previous claims were incorrect or confusing. We amended the entanglement swapping section in the supplementary material where we corrected mistakes and removed ambiguous statements. We believe these revisions improve the clarity of presentations and accuracy of our results. We hope the reviewer would be satisfied with the changes and our response.

The authors have provided a comprehensive reply to my previous comments and have made significant changes and amendments to the manuscript. However, the added material triggers some additional questions and raises doubts about the validity of some parts of the analysis and conclusions presented by the authors. In what follows, I will focus on the discussion of entanglement swapping presented in Section V of Supplementary material.

The authors claim that using the heralded teleporter they can outperform standard deterministic CV entanglement swapping schemes. The considered protocol is illustrated in Fig. 8 of Supplementary Material. Let us first consider a deterministic entanglement swapping. Assume that the two shared entangled states are two-mode squeezed vacuum states with squeezing constants r_1 and r_2 , respectively. The input state is a pure Gaussian state and the two-mode homodyne detection in the CV entanglement swapping is a Gaussian measurement. Therefore, for each particular measurement outcome, the state of Alice and Bob will be a pure Gaussian state, in particular a coherently displaced two-mode squeezed state. Note that the covariance matrix of the conditional state is fixed and does not depend on the measurement outcome, only the coherent displacement is a function of the measurement outcome. Alice and Bob can undo this displacement to deterministically recover a two-mode squeezed vacuum state with squeezing constant $\tanh(r) = \tanh(r_1)\tanh(r_2)$. Note that the optimal corrective displacement generally need not be unity-gain and the feedforward rescaling factors should be optimized depending on r_1 and r_2 .

Let us now switch to the probabilistic heralded protocol. This protocol will filter the ensemble of conditional states, i.e. it will modify the probabilities of each conditional state (that corresponds to a specific outcome of homodyne detection). However, all these conditional states have the same covariance matrix M and they differ only by coherent displacements. Therefore, covariance matrix G of any statistical mixture of such states will satisfy $G \geq M$, hence the post-selection and filtering cannot improve the entanglement.

There is also another argument why the filtering proposed by the authors should not work, and this is based on the no-go theorems for entanglement distillation of Gaussian states with Gaussian operations. Let $r_1 > r_2$. It appears that the authors claim that their scheme can achieve entanglement swapping where the output squeezing is equal to r_1 , i.e. $r_{out} = r_1$. Suppose now that Alice and Bob share the two-mode squeezed vacuum with parameter r_2 . Alice locally prepares the two-mode squeezed vacuum state with parameter r_1 and then Alice and Bob apply the heralded protocol. Using local Gaussian operations, classical communication and conditioning they would probabilistically increase the entanglement shared between Alice and Bob, which is known to be impossible.

The above discussion suggests that the analysis presented by the authors may not be completely reliable and therefore the manuscript is not suitable for publication.

We apologize for a mistake we made in the added supplementary analysis for teleporting EPR input states. And we appreciate the reviewer for pointing this out. In the present supplementary note, we have corrected the note and verified the remaining results using multiple approaches.

The reviewer is correct that the conditional states represent the optimal swapping output in terms of entanglement, EPR correlation and the output purity [PRA 83, 012319]. Our analysis aligns with this statement. By employing two-sided displacements as well as optimized feedforward gains, the deterministic entanglement swapping scheme can obtain the same results as the conditional states. In this regard, we agree that any post-selection is not able to surpass the single-shot swapping or the optimal ensemble-average swapping based on two-sided displacements.

However, we would like to emphasize that we propose and experimentally demonstrate here a novel quantum teleportation scheme that improves upon conventional teleportation schemes [PRL 80, 869] – not entanglement swapping, and we report the world-record fidelities using the proposed technique. We agree with the reviewer that it is interesting to explore how the teleporter behaves for EPR input states in comparison to the deterministic conventional teleporter in this regard. The supplementary note was added purely for this purpose. It is to us irrelevant and unfair to compare our teleporter with the optimal entanglement swapping protocol as the two schemes are designed for different purposes and key distinctions exist between the two:

1. The operational regimes of interest for entanglement swapping are considerably different. For example, while unity gain is crucial in quantum teleportation, it is not the optimal operating point for entanglement swapping.
2. The setups are different: optimal swapping would require two-sided displacements, whereas quantum teleportation involves only one-sided displacements.
3. The performance measures are often different. Whilst fidelity is the commonly used figure of merit for quantum teleportation, it is not normally used in evaluating entanglement swapping, where purity, entanglement are more relevant measures.

Although the optimal entanglement swapping offers better performance than our heralded teleporter, it would not be suitable for teleporting more general input states. As an example, for arbitrary coherent input states, our teleporter would achieve significantly higher fidelity as compared to the swapping scheme when benchmarked on the same EPR resource. Nevertheless, as shown in the supplementary note, our teleporter indeed performs comparably with the optimal swapping (either conditioned on single outcome at the Bell measurement or using two-sided displacements) in terms of both the entanglement and purity with a reasonable success probability. The results reinforce the fact that our teleporter provides a versatile tool that improves significantly upon conventional teleportation schemes by trading determinism. As pointed out by the second reviewer and quoted here, *“the technique can be seen as a bridge between the deterministic teleportation based on a feed-forward and probabilistic teleportation based on post-selection with the added benefit of not enforcing the attenuation that the probabilistic teleportation requires. In this sense, the presented method naturally expands the existing toolbox of quantum optics experiments.”*

On the other hand, a more relevant comparison should be between our heralded teleportation and the conventional deterministic teleportation used for teleporting EPR states as those presented in [PRA 61, 010302, PRA 60, 2752]. We summarize in the following the preliminary simulation results for the performance comparison.

We show that in an ideal transmission channel, the heralded teleporter is capable of preserving the purity of the input states effectively, while deterministic teleportation tends to produce highly mixed outputs. The results are summarized in the following figure which highlights the potential for achieving higher purity by sacrificing the success rate of the teleportation process.

Fig.1 Entanglement of formation and purity as a function of the input EPR squeezing in an ideal transmission channel. Both the input EPR and the channel EPR are pure. The red dashed curve represents the conditional state (single-shot output) that corresponds to the optimal achievable swapping purity and entanglement of formation, and hence serves as an ultimate limit on the performance of both the heralded and the deterministic teleporters. Note that the heralded teleporter exhibits a significant enhancement in the purity of the output two-mode states as compared to the deterministic teleportation scheme with optimized rescaling factors.

On the other hand, in the presence of channel loss where the loss is imposed on the submode of the channel EPR sent to the Bell measurement, the heralded teleporter achieves significant enhancement in both entanglement and purity, when compared to deterministic teleportation. In both scenarios, the heralded teleporter performs comparably with the optimal entanglement swapping schemes. Despite the loss, the heralded teleporter yields two-mode output states with comparable purity and EOF to the conditional state. In contrast, the output of the deterministic teleportation is highly mixed, indicating its susceptibility to the negative effects of transmission losses.

Fig.2 Entanglement of formation and purity as a function of the input EPR squeezing in a pure lossy channel. Here the input EPR squeezing r_1 and the EPR resource squeezing r_2 are equal ($r_1 = r_2 = r$). The heralded teleporter

achieves significant enhancement in entanglement as compared to the conventional deterministic teleporter based on one-sided displacements. The conditional state (single-shot output) is shown in red-dashed curve as an ultimate benchmark.

In summary, by including these additional results, we have provided further insights into the performance comparison between the heralded teleporter and deterministic teleportation in the specific context of EPR teleportation, particularly in nonideal transmission channels. We have corrected the note and verified the validity of the presented results using multiple approaches. By demonstrating consistency with established formulas and previously published results, we reinforce the robustness and accuracy of our analysis approach. We hope the reviewer would be satisfied with our response and the updated supplementary note.

Reviewer #2

We thank the reviewer for the thorough appraisal and positive response regarding our paper. We appreciate the reviewer's constructive comments that highlight the importance of our results.

I would like to thank the authors for revising and expanding the manuscript. Showing that the technique can, in principle, work even for non-Gaussian and entangled states significantly contributes to its usability and possible impact.

I believe the technique can be seen as a bridge between the deterministic teleportation based on a feed-forward and probabilistic teleportation based on post-selection with the added benefit of not enforcing the attenuation that the probabilistic teleportation requires. In this sense, the presented method naturally expands the existing toolbox of quantum optics experiments and will most likely see further use or at least testing.

I therefore recommend the manuscript for publication.

If I may have one final recommendation, I would suggest that, in the places of the manuscript where the new technique is compared to the deterministic teleportation, authors also compare it to the probabilistic teleportation based on post-selecting on a single value of homodyne measurement. For some scenarios, such as teleportation of a photon or entanglement swapping, this approach provides significantly better results than deterministic teleportation and it would benefit the reader considerably to see a full and fair comparison with the state-of-the-art.

We appreciate the reviewer's suggestion to compare our work with the post-selecting scheme based on a single value of the homodyne measurement. It is indeed an interesting avenue to explore. The reviewer is correct that conditioning on a single outcome at the continuous-variable Bell measurement could lead to significantly better results than deterministic teleportation in certain scenarios, such as for single photon inputs and entanglement swapping. We provide some preliminary results and discussions in this regard in the Supplementary note.

Nevertheless, it is worthwhile stressing that although conditioning on single outcome can lead to enhanced performance in certain scenarios, these advantages may not be applicable to general input states. For example, coherent states with large quadrature amplitudes may not benefit from the single-shot scheme as much as our heralded teleporter, which can still achieve significant fidelity enhancement in such cases. Moreover, our heralded teleporter demonstrates comparable performance to the single-shot scheme in the context of swapping with reasonable success probabilities. We acknowledge that this topic merits further thorough investigations which we intend to explore for future studies.

Reviewer #3

We thank the reviewer for the time and effort in reviewing our manuscript. We are glad that the reviewer finds the revisions satisfactory, and all his concerns were adequately addressed. We appreciate the reviewer's constructive comments.

The authors made an impressive effort to address all the reviewers' comments and went beyond what would usually be expected in terms of clarifications and new material in the revised version. They clearly addressed all my comments. The paper is significantly improved with respect to readability and impact, and I can now surely recommend publication in NatComm.

We thank the reviewer for the affirmation and his support.

REVIEWERS' COMMENTS

Reviewer #1 (Remarks to the Author):

The authors have made the necessary changes and amendments and they corrected the part of the Supplementary Material that is devoted to teleportation of EPR states. The present analysis and discussion of the teleportation of EPR states appears to be sound. The manuscript proposes an intriguing concept to conditionally improve the fidelity of quantum teleportation and this proposal is experimentally demonstrated and tested with coherent and thermal states. The results are of sufficient novelty and potential interest to deserve publication.

Reviewer #1

The authors have made the necessary changes and amendments and they corrected the part of the Supplementary Material that is devoted to teleportation of EPR states. The present analysis and discussion of the teleportation of EPR states appears to be sound. The manuscript proposes an intriguing concept to conditionally improve the fidelity of quantum teleportation and this proposal is experimentally demonstrated and tested with coherent and thermal states. The results are of sufficient novelty and potential interest to deserve publication.

We are glad that the reviewer finds the revisions satisfactory and the new analysis for teleporting EPR states sound. We appreciate that the reviewer finds our results intriguing and of sufficient novelty. We thank the reviewer for the constructive comments and support for publication.